



# Hydrologic Flowpath Development Varies by Aspect during Spring Snowmelt in Complex Subalpine Terrain

Ryan W. Webb[1], Steven R. Fassnacht[2], Michael N. Gooseff[1]

[1]Institute of Arctic and Alpine Research, University of Colorado, Boulder, CO, 80309, United States

[2]Department of Ecosystem Science and Sustainability, Colorado State University, Fort Collins, CO, 80523, United States

*Correspondence to*: Ryan W. Webb (ryan.w.webb@colorado.edu)

**Abstract.** In many mountainous regions around the world, snow and soil moisture are key components of the hydrologic cycle. Preferential flowpaths of snowmelt water through snow have been known to occur for years with

few studies observing the effect on soil moisture. In this study, statistical analysis of the topographical and hydrological controls on the spatio-temporal variability of snow water equivalent and soil moisture during snowmelt was undertaken at a subalpine forested setting with north, south, and flat aspects as a seasonally persistent snowpack melts. We investigated if preferential flowpaths in snow can be observed and the effect on soil moisture through measurements of snow water equivalent and near surface soil moisture in addition to observing how SWE and near

surface soil moisture vary on hillslopes relative to the toes of hillslopes and flat areas. We then compared snowmelt infiltration beyond the near surface soil between flat and sloping terrain during the entire snowmelt season using soil moisture sensor profiles. This study was conducted during varying snowmelt seasons representing above normal, relatively normal, and below normal snow seasons in northern Colorado. Evidence is presented of preferential meltwater flowpaths at the snow-soil interface on the north facing slope with the effects observed in changes in SWE

and infiltration into the soil at 20 cm depth; less association is observed in the near surface soil moisture (top 7 cm). We present a conceptualization of the meltwater flowpaths that develop based on slope aspect and soil properties. The resulting flowpaths are shown to increase the snow water equivalent by as much as 170% at the base of a north facing hillslope. Results from this study show that snow acts as an extension of the vadose zone during spring snowmelt and future hydrologic investigations will benefit from studying the snow and soil together.

**1 Introduction**

In many mountainous headwater catchments snow and soil moisture are key components of the hydrologic cycle, providing valuable information pertaining to the dynamic processes that occur during spring runoff. This has justified large data collection efforts to further understand the distribution of snow and soil moisture across landscapes during the winter and spring seasons (Elder et al., 2009). During spring, much of snowmelt will infiltrate into the soil with a

noticeable change in soil moisture prior to recharging groundwater storage, producing streamflow, or contributing to evapotranspiration (Bales et al., 2011; Kampf et al., 2015; Webb et al., 2015). The relative saturation in the vadose zone controls the stream connectivity and release of water and nutrients from subsurface storage into stream systems (McNamara et al., 2005; Williams et al., 2009b). Soil moisture during this time is driven by snowmelt that can impact the water availability for plant production (Molotch et al., 2009; Harpold et al., 2015) as well as the ionic signature of

soil moisture and stream flow (Harrington and Bales, 1998). For these reasons the connections between snowmelt and





soil moisture are critical in understanding the hydrologic cycle in snow dominated headwater systems (Jencso et al., 2009), particularly in the face of a changing climate that will alter the snowmelt season and resulting hydrological dynamics (Adam et al., 2009; Clow, 2010; Clilverd et al., 2011; Harpold et al., 2012; Rasmussen et al., 2014; Fassnacht et al., 2016).

Processes within headwater catchments such as snow accumulation and persistence are known to vary at multiple scales of interest. From a basin scale perspective, elevation has been shown to influence the depth and persistence of a snowpack (Richer et al., 2013; Molotch and Meromy, 2014; Sexstone and Fassnacht, 2014) while at finer resolutions the spatial variability of both accumulation and melt may be controlled by aspect (Williams et al., 2009a; López-Moreno et al., 2013; Hinckley et al., 2014) and snow in forested areas are affected by interception during accumulation,

shortwave radiation shading, and longwave radiation influences prior to and during melt (Storck et al., 2002; Musselman et al., 2008; Molotch et al., 2009; Adams et al., 2011; Webb, in review). However, far less is known about the variability that snowmelt has on soil moisture and flowpaths during snowmelt at the hillslope scale, in large part due to the difficulty of observing soil moisture beneath a deep snowpack at high spatial resolution. Snowmelt is important to soil moisture storage and resulting streamflow (McNamara et al., 2005; Williams et al., 2009a; Bales et

al., 2011; Hunsaker et al., 2012; Kormos et al., 2014). Stream connectivity to the surrounding landscape follows seasonal trends with the highest connectivity during spring snowmelt based on factors such as topography (McNamara et al., 2005; Jencso et al., 2009; Jencso and McGlynn, 2011). The aspect of a hillslope will additionally increase soil water storage and retention on north aspect slopes (Geroy et al., 2011) that can alter runoff processes and result in spatially variable soil moisture beneath a melting snowpack.

The ability to observe soil moisture throughout the water year has seen recent advancements for capturing high resolution data at both spatial and temporal scales (e.g. Bales et al., 2011). Similar advances have occurred for observing variables such as the liquid water content of a snowpack (Mitterer et al., 2011; Techel and Pielmeier, 2011; Koch et al., 2014; Heilig et al., 2015). This has allowed for further understanding of hydrological systems and dynamic processes that are vulnerable to climate change (Bales et al., 2006). However, observations of the relative saturation

of soil beneath a snowpack has been limited to an array of discrete points with sufficient instrumentation, and few studies have investigated spring snowmelt soil moisture at the similar scale as the snow above it has been measured. The few studies that have observed these process have shown microtopography to influence infiltration across the snow-soil-interface (SSI) (French and Binley, 2004) and that wetter areas tend to remain wetter with slope and aspect being important factors at a low elevation site (Williams et al. 2009a). In high elevation alpine environments

topography and wind shielding influences soil moisture distribution though there is less association to these parameters in low snow years (Litaor et al., 2008). These studies, limited to high alpine and low rain-snow transition zones, suggest that topographic influences on soil moisture are strong but more investigations during varying snow accumulation, melt dynamics, and environments are important, particularly with variable regional and environmental snowpack responses to climate variability (Harpold et al., 2012), to connect the distribution of soil moisture across a

landscape to runoff processes.

The relative saturation of the vadose zone determines runoff processes during spring snowmelt (McNamara et al., 2005). Runoff processes have been shown to change during spring snowmelt compared to summer rain events



(Eiriksson et al., 2013; Williams et al., 2015). During snowmelt, soil moisture is influenced most in the top 10 cm of soil (Blankinship et al., 2014) with pulses of water that reach further depths varying widely at both the hillslope and catchment scale (Webb et al., 2015). At the catchment scale a south aspect hillslope may display matrix flow during snowmelt as the north aspect displays evidence of preferential flow through the soil (Hinckley et al., 2014).

Preferential flowpaths have been shown to occur both in the soil beneath a snowpack (French and Binley, 2004) and above the ground surface within the snowpack (Marsh and Woo, 1985; Kattelmann and Dozier, 1999; Williams et al., 2000; Liu et al., 2004; Williams et al., 2010). Preferential flow within a snowpack can form as the result of ice lenses (Colbeck, 1979) or differences in grain size and density (Webb et al., in review). Each of these can alter the flow of water through snow and resulting infiltration into the soil from the centimeter scale (Williams et al., 2010) up to tens

of meters (Eiriksson et al., 2013; Webb et al., in review). Preferential flowpaths within a snowpack will create spatially variable snowmelt patterns across a landscape depending on the variable metamorphism that occurs within the snowpack (Yamaguchi et al., 2010; Adams et al., 2011; Domine et al., 2013; Katsushima et al., 2013), which increases during melt (Marsh, 1987). These melt patterns have been shown to have correlation lengths of five to seven meters in relatively flat alpine terrain (Sommerfeld et al., 1994; Williams et al.,1999) and lesser correlation lengths of two to

four meters in subalpine terrain (Webb, in review). Preferential flowpaths within a snowpack will alter soil moisture and resulting runoff processes at the hillslope and catchment scales.

To our knowledge there has not been a study investigating snow and soil moisture interactions specifically to investigate hydrologic flowpath development in a sub-alpine environment beneath a deep (2 m) seasonally persistent snowpack. The goal of this study is to gain further understanding through observations of flowpath development in a

snowmelt dominated subalpine headwater catchment. Observations of near surface soil volumetric water content (VWC) were compared to topographical parameters (e.g. slope, aspect, etc.) and hydrological variables (e.g. temperature, date of peak SWE, etc.). Statistical analysis of the topographical and hydrological controls on the spatio-temporal variability of snow and soil moisture during snowmelt was undertaken at a subalpine forested setting with north, south, and flat aspects as a seasonally persistent snowpack melts through the following objectives: 1) investigate

if preferential flowpaths in snow can be observed and the effect on soil moisture through measurements of snow water equivalent (SWE) and near surface soil moisture, 2) observe how SWE and near surface soil moisture vary on hillslopes relative to the toes of hillslopes and flat areas, and 3) compare snowmelt infiltration beyond the near surface soil between flat and sloping terrain during the entire snowmelt season.

## 2 Methods

To understand flowpath development during snowmelt and the resulting distribution of soil moisture, observations of SWE and near surface soil moisture were correlated to test the influence of topography and snow on soil moisture using Pearson's correlation coefficient, r, and a level of significance determined at p-values of 0.05 and 0.01. Near surface soil volumetric water content (VWC) was compared to SWE at the same location on the date of observations, SWE on the first survey date (representative of peak SWE), the change in SWE between survey dates prior to

measurement, near surface VWC on the first survey date, and topographic slope, elevation, and northness as calculated

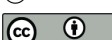



from the 10 m digital elevation model (DEM) (USGS, 2015). Northness is defined as the product of the cosine of aspect and the sine of slope (Molotch et al., 2005; Sexstone and Fassnacht, 2014).

### 2.1 Study Site

Data were collected over a 0.2 km$^2$ area near Dry Lake in Routt National Forest, approximately 6.5 km northeast of
Steamboat Springs, Colorado (Fig. 1b). The elevation of measurement locations ranged from 2500 m to 2600 m with slope angles from 1° to 30° as determined from a 10 m DEM (USGS, 2015). The site has a mix of deciduous (aspen, populous tremuooides) and evergreen forest (subalpine fir, Abies lasiocarpa and engelman spruce, Picea engelmanii) with a majority of the vegetation growing near the small stream and large areas of open canopy conditions (Fig. 1a) on each of the two predominant hillslopes (one south-southeast facing, and one north-northwest facing).
The soils are primarily loams with very cobbly loam dominating the south aspect slope, cobbly sandy loam on the north aspect, and loam on the flatter aspects with observations of highly organic soils in the flat northeastern section of the area at the base of the north aspect hillslope (Table 1). Depth to bedrock was estimated using a one meter long hand auger at 16 locations within the study site along a transect from the top of the south aspect slope to the top of the north aspect slope (Fig. 1b), resulting in soil depths ranging from 12 cm to greater than one meter at a single location.
Soil depths tend to decrease with increasing elevation with a mean depth to bedrock of 40 cm and a median of 38 cm calculated from the 15 depths less than one meter (Fig. 1c). Sieve analyses were also conducted on six different volumetric samples of approximately 200 cm$^3$ for near surface soil collected from four locations (two locations sampled twice) (Table 1).

For this study, regions were defined to compare observations on varying aspects and at the toes of hillslopes. Regions
were defined as middle of the south aspect slope (SM), toe of the south aspect slope (ST), flat aspect (FA), toe of the north aspect slope (NT), low on the north aspect slope (NL), and high on the north aspect slope (NH) (Fig. 1c).

### 2.2 Spatial Surveys

Spatial surveys were conducted in 2013, 2014, and 2015. In 2013 two surveys were conducted four weeks apart while in 2014 and 2015 four surveys were conducted at two week intervals. All survey periods began during the first week
of April (April 6, 2013; April 4, 2014; April 3, 2015). Surveys consisted of a series of snow pits for collecting near surface soil moisture and bulk SWE data. At each pit location, the first measurements taken were near surface soil moisture using a handheld time domain reflectometer (TDR) (FieldScout TDR 100; Spectrum Technologies, Inc.) to measure the VWC using seven centimeter long prongs inserted vertically into the soil. A total of five TDR measurements were averaged across the bottom of each snow pit (approximately one meter across, measurements ~20
cm apart). Volumetric soil samples (~40 cm$^3$) were collected at three of the same point locations as TDR measurements in each snow pit for laboratory confirmation of VWC during surveys in 2013 and 2014. Bulk SWE measurements were collected using a plastic tube with an inner diameter of 68 mm and a length of 1.8 m. A core was collected for the full depth of the snowpack when possible, and in no more than two segments when the depth of the snowpack was greater than the length of the tube. Snow cores were placed in a plastic bucket and mass measured using a digital scale
with 10 g precision. Two cores were averaged per snow pit with additional measurements if the first two showed



greater than ten percent mass difference (a rare occurrence). When locations were returned to a new pit was dug within one to two meters with care to avoid previously disturbed snow.

On April 6, 2013, 15 snow pits were measured and six were returned to and measured again on May 4, 2013 to capture the changes at the SM, ST, FA, NL, and NH regions (Webb and Fassnacht, 2016a). The 2014 and 2015 surveys

collected data along approximate north-to-south transects perpendicular to topographic contours collecting multiple measurements in the six regions of interest. In 2014 a total of 25 snow pits were measured on April 4 and nine of these pits were returned to in two week intervals through May 17; eight of the nine pits were measured on April 19 (Webb and Fassnacht, 2016b). The 2015 surveys made observations at 47 snow pits on April 3 and 23 of these pits were returned to on two week intervals through May 16 (Webb and Fassnacht, 2016c). Snow pit measurements were then

averaged at each of the regions (SM, ST, FA, NL, and NH) for each day of observations.

### 2.3 Time Series Data

At the study site, there are two stations that measure meteorological variables including air temperature, relative humidity, precipitation, wind speed and direction, and solar radiation. The Dry Lake Remote Automated Weather Station (RAWS station coCDRY and National Weather Service ID 050207, <raws.dri.edu>) is along an exposed ridge

at the top of the south aspect slope at approximately 2540 m elevation and has been operated by the United States Forest Service since 1985 (Fig. 1a and 1b). Additionally, hourly dewpoint and wet bulb temperature, snow depth, and SWE are measured at the Dry Lake Snow Telemetry (SNOTEL, station 457 <wcc.nrcs.usda.gov>) station located approximately 120 m to the south-southwest of the RAWS at a lower elevation of 2510 m with light canopy shading (Fig. 1a) and has been operated by the Natural Resources Conservation Service since 1980 measuring SWE and

precipitation. Since 2003, this SNOTEL station has additionally measured soil moisture and temperature at three depths (5 cm, 20 cm, and 50 cm). The RAWS and SNOTEL data provide meteorological data at two elevations and different canopy conditions within the relatively small area of interest of this study.

Snowmelt infiltration observed by the SNOTEL station is for a relatively flat location, and thus was compared to additional soil moisture and temperature instruments that were installed at a single location on the north aspect slope

on December 27, 2013 at depths of 5 cm, 12.5 cm, and 20 cm. The top and bottom depths match two of the SNOTEL soil moisture depths; the 12.5 cm depth sensor was added at mid-depth between the other sensors. Instruments installed were Decagon Devices, Inc. 5TM temperature and moisture sensors connected to a Decagon Em50 data logger. Installation in December 2013 required disturbing the snowpack and soil, thus the snowpack and soil moisture were allowed to return to near undisturbed conditions after installation and data prior to March 15, 2014 was not included

in analysis. The soil moisture sensors and data logger were calibrated prior to installation using approximately 1500 cm$^3$ of soil collected from the study site and tamped around a sensor to a density of 1.0 g cm$^{-3}$, similar to measured conditions in the field. The calibration occurred at a constant temperature of 0.5°C and additions of seven to ten percent VWC every four days. The container mass was recorded to confirm mass of soil, sensor, and water as well as the sensor reading of temperature and VWC prior to the addition of water each time. All mass recordings were at a

precision of 1.0 g (volumetric water precision of 0.06%) and VWC sensor recordings to 0.1%.





## 3 Results

### 3.1 Time Series Snow and Meteorological Data

The three spring snowmelt seasons studied represent varying melt conditions. Average peak SWE occurs at the Dry Lake SNOTEL station on April 5 with a 35 year median peak of 570 mm and a mean of 590 mm (Fig. 2a). Peak SWE

values recorded at the SNOTEL station were 495 mm, 715 mm, and 415 mm for 2013, 2014, and 2015, respectively, representing 87%, 125%, and 73% of the station long term median. Peak SWE timing ranged from March 9 in 2015, preceding first survey by nearly one month, to April 25 in 2013, 19 days after the first survey (Fig. 2a). The number of days from peak SWE no snow recorded at the SNOTEL station ranged from 22 days in 2013 to 52 days in 2015, with each year having of incremental snowfall during the melt period (Fig. 2b). Precipitation at the SNOTEL station

during the survey periods was 130 mm for 2013, 100 mm for 2014, and 2015 accumulated 115 mm from the date of the first survey to the last survey (Fig. 2b). The precipitation that fell during the melt period in 2015 likely included a number of rain-on-snow events due to the regular warmer than freezing temperatures in late April and May (Fig. 2d), but snow can fall at several degrees warmer than zero (Fassnacht et al., 2013). On March 1st, the snow accumulation was the same in 2013 and 2015, with approximately 40% more in 2014; the subsequent spring snowpack variability

between years was a result of varying meteorological forcing conditions during March, April, and May (Fig. 2). The SNOTEL station data show air temperature during these months warmer than freezing 62%, 64%, and 77% of the time and cumulative solar radiation totaled 355 kW, 380 kW, and 400 kW in 2013, 2014, and 2015, respectively (Fig. 2c). Wind directions remained consistent each year during the spring months, generally from the southwest and northeast alternating diurnally between directions (Fig. 2e). The RAWS site showed slightly larger diurnal temperature

fluctuations, greater cumulative solar radiation, and less precipitation during the spring snowmelt seasons relative to the SNOTEL station, though generally similar conditions were observed when comparing the two stations (Fig. 2).

### 3.2 Spatial Surveys

Spatial surveys conducted in 2013 and 2014 occurred while a measureable snowpack was still observed at the SNOTEL station for all survey dates whereas in 2015 the SNOTEL station measured zero snow for the final two of

the four surveys resulting in variable conditions for SWE and VWC measurements each year. In 2013 all north aspect locations increased in SWE between surveys with the largest increase occurring at the toe of the north aspect slope (NT, 160 mm) and the smallest increase high on the slope (NH, 20 mm) (Fig. 3bi). SWE also increased at the toe of the south aspect slope (ST, 90 mm) and decreased in the middle of the south facing slope (SM) (Fig. 3bi). In 2014 a similar pattern was observed of increases in SWE at the toes of each slope (ST and NT) and lesser increases on the

north facing hillslope (Fig. 3biii), though low on the slope one pit location decreased in SWE and another remained the same (Fig. 3aiii). Increases in SWE occurred early in the melt period for 2013 and 2014 whereas the early melt period was not observed in 2015 due to the early peak accumulation. After the initial increase in SWE for some locations during the first melt period observed (MP-1), all locations decreased in SWE for the two following melt periods observed (Fig. 3b). In 2015, SWE did not change during MP-1 at NT and ST regions, while it decreased at the

four other locations (Fig. 3bv). This is less of a decline in regions that increased for 2013 and 2014. At the toes of each slope in 2014, the increase in SWE during the first melt period was larger than the decrease in the following two



melt periods combined (Fig. 3iii). In 2015, only two measurement locations had snow during the final survey (May 16) and precipitation influenced observations (discussed later).

Also observed during manual surveys in 2013 and 2014 were the presence of frozen ice "veins" immediately above the snow-soil interface (SSI) (Fig. 4). These were observed on the north aspect (NL and NH) and at the toe of the

north aspect slope (NT) only and appeared to be continuous. The occurrence of this phenomenon was in the direction of the hillslope fall line and on ground that was not super-saturated. These ice "veins" were not observed in 2015. Also qualitatively observed was the relative density of snow in each pit in 2014. On the north aspect slope, snow density tended to decrease with height above the SSI, with heavy wetter snow remaining in the bottom of the snowpack and the thickness of the higher density snow at the bottom of each snow pit increasing downslope.

The near surface soil VWC during all surveys varied from mean values of 15% to 85% (Fig. 3). The maximum soil moisture consistently occurred at the toe of the north aspect hillslope (NT) in the highly organic soil. The soil at this location was also observed to be super-saturated during a single survey on April 19, 2014 (resulting in the 85% VWC, Fig. 3bii). The near surface soil VWC showed variable observations of increasing and decreasing soil moisture beneath a melting snowpack with relatively larger decreases immediately following snow disappearance (Fig. 3a).

The 2015 surveys resulted in the largest variability of measurements each survey for both SWE and near surface VWC (Fig. 3c). The early peak accumulation resulted in only two measurement locations with snow for all four survey dates. However, as with previous years, near surface VWC decreased noticeably after the disappearance of snow for all locations with some increases due to rain events (Fig. 3cii).

### 3.3 Spatial Correlation

The topographic parameters of elevation, slope, and northness showed mostly low correlations to near surface VWC during observations and little significance at the 0.05 level (Table 2). The only topographic parameter that resulted in a Pearson's r value of magnitude larger than 0.5 was slope, occurring later in the 2014 observation period. The only topographic parameter that showed any significance was northness on May 16, 2015, when soil had been exposed to the atmosphere from loss of snow for 93% of measurement locations.

The hydrologic variables tested for correlation to near surface VWC included SWE, ΔSWE, first measured SWE, and first measured VWC. These variables showed higher correlations and more occurrences of significance at the 0.05 and 0.01 level relative to topographic parameters (Table 2). The highest Pearson's r values of all variables to near surface VWC was the first measured near surface VWC that are positive and all but one correlation being significant at the 0.01 level. Pearson's r values tend to decrease in magnitude for this variable as time from the first survey

increases (Table 2).

The correlation of SWE variables to near surface VWC were inconsistent in strength, direction, and significance with a lot of variability each survey. The mostly negative correlations for near surface VWC to ΔSWE indicate that in 2013 and 2014 locations with lesser changes in SWE had higher near surface VWC, though this was significant at the 0.05 level on April 19, 2014 only (Table 2). The similar negative correlation and magnitudes of near surface VWC to first

measured SWE show that in 2013 and 2014 areas that had less SWE during the first survey tended to have higher measured VWC, significant at the 0.05 level on April 19, 2014, and at the 0.01 level on May 3, 2014 (Table 2).



### 3.4 VWC Time Series Data

Soil moisture and temperature sensors clearly show the diurnal fluctuation of VWC from snowmelt infiltration across the SSI and the fluctuation in soil temperature as snow disappears (Fig. 5). Soil temperatures at 5 cm depth remain between $0^0C$ and $1^0C$ through winter and temperatures begin to fluctuate in the soil at approximately the same time of snow disappearance (Fig. 5). This temporal pattern occurs at both the SNOTEL station and on the north aspect slope. These locations also show the relatively quick drying after snow disappearance in 2014 and slower drying as a result of rain in 2015. However, there is more drying between rain events on the flat aspect (Fig. 5bi) compared to the north aspect slope (Fig. 5bii). At the flat aspect SNOTEL station the VWC sensors at 5 cm and 20 cm depths follow a similar temporal pattern remaining within 5% of each other the entire winter season indicating snowmelt infiltrating and wetting the soil at 20 cm depths and a higher relative saturation in entire vadose zone (Fig. 5bi). Beneath the snowpack and during melt, the north aspect hillslope VWC sensors show a difference of approximately 15-20% with more similar VWC observed during summer and fall rain events (Fig. 5bii). The 12.5 cm deep sensors on the north aspect slope also displays a more similar VWC value to the 20 cm sensor with values drier than the 5 cm sensor indicating less snowmelt infiltrating and wetting the soil at 12.5 cm and 20 cm depths, and a lesser relative saturation in the vadose zone on the north facing slope (Fig. 5bii) compared to flat terrain (Fig. 5bi).

Rain events that occurred prior to soil moisture drying in May, 2015 resulted in infiltration excess overland flow due to high intensity precipitation. These events occurred prior to new vegetation becoming established on the hillslope. Evidence of overland flow was observed during the May 16, 2015 survey when most of the snow had disappeared on all hill slopes and the dead grasses from the previous summer were lying flat on the ground in the downslope direction; this was not the observed state of the dead grasses in snow free areas during the previous survey on May 2. During the overland flow event(s), differences in VWC measurements at 5 cm, 12.5 cm, and 20 cm deep sensors on the north aspect slope are similar to what is observed during the snowmelt season and not what is observed during rain events during the summer and fall (Fig. 5bii). However, the flat aspect VWC sensors displayed similar patterns during nearly all rainfall or snowmelt events (Fig. 5bi). These observations indicate that less snowmelt infiltrates to the 20 cm depth on the north facing slope relative to the flat aspect (Fig. 5b) and that snowmelt water is flowing downslope at the SSI (Fig. 4).

## 4 Discussion

The multiple years of observation at a subalpine location with a deep seasonally persistent snowpack offers analysis of SWE and near surface VWC patterns that have previously been limited to lower elevations near the rain-snow transition zone (Williams et al., 2009a) and higher elevations in an alpine environment (Litaor et al., 2008). In this study, the only topographic parameter that displayed any significance on the near surface soil moisture at the 0.05 level was northness and this appeared to increase in significance and strength with time indicating that it is likely more related to the presence or absence of snow and influences from rain (Table 2). However, infiltration of snowmelt beneath the near surface to 12.5 cm or 20 cm depth was influenced by slope with more infiltration wetting the soils at 20 cm depth on the flat aspect and lesser wetting at this depth on the north aspect (Fig. 5b) where observations of ice




"veins" were made at the SSI (Fig. 4). The soils on the south aspect slope are generally coarser than the north or flat aspects (Table 1), suggesting that soil water retention is higher on north aspects (Geroy et al., 2011). This was reflected with the north aspect soils having higher water contents than south aspect soils, both with and without the presence of a snowpack (Fig. 3). Near surface soil moisture approached relative equilibrium with a persistent snowpack rather

than varying through time (Fig. 3, Fig. 5b). As the snowpack melts the shallow subsurface VWC displays clear diurnal fluctuations (Fig. 5b). The locations that were wet relative to the other locations during the first survey remained as such for all following surveys when snow persistently covered the study area comparing well to the study by Williams et al. (2009a) at a smaller scale and beneath a shallower snowpack at a lower elevation. However, in contrast to Williams et al. (2009a), near surface VWC showed a negative correlation to the first measured SWE (representative

of peak) in 2013 and 2014 resulting from locations with deeper snowpacks having lesser near surface VWC and shallower snowpacks having greater near surface VWC. This is the result of the shallower snowpacks during the first survey being near the bottom of the slopes and in the flat terrain influenced by canopy interception (Fig. 1a) that yields a lower peak SWE followed by melt flowing downslope at the SSI towards these locations and increasing both SWE and near surface VWC (Fig. 4). Though in 2015, a relatively low snow year, results agreed with Williams et al. (2009a)

of higher near surface VWC at locations that accumulated more snow indicating the amount of snowfall is also important to these processes. During low snow years, areas where snow persists longer will result in a longer influence on near surface soil moisture. Near surface VWC will additionally depend on variability in soil parameters such as soil water retention, with higher moisture retention from finer soil particles, similar to the north aspect slope, will affect the infiltration or lateral flow of meltwater at the SSI when on a slope.

Meltwater flowing downslope at the SSI on the north aspect hillslope is shown by the increases in SWE at locations on and at the toe of the hillslope (Fig. 3), the frozen "ice veins" observed (Fig. 4), less infiltration to 12.5 cm and 20 cm depth on the slope (Fig. 5), similarities in soil moisture between snowmelt and overland flow rain events, and the qualitative observations of snow density and wetness increasing with depth in each snow pit. The movement of water across layer interfaces has been shown within a snowpack (Williams et al., 2000; Liu et al., 2004; Williams et al.,

2010) and at the SSI (Eiriksson et al., 2013), with the latter being observed in this study (Fig. 4). This phenomenon will depend on soil parameters, snowpack layer characteristics, slope angle, and the rate that meltwater is percolating through the snowpack. These factors will determine if an interface acts as a permeability barrier, similar to a soil drain, or a capillary barrier (Webb et al., in review). Flow at the SSI was not observed on the south slope though SWE did increase at the toe (ST, Fig. 3). However, here there is much local terrain variability, and snow depth also increased

with SWE at the toe of the slope (ST) resulting in less certainty in meltwater flow near the SSI as the cause of SWE increase. Furthermore, this hillslope has coarser soil and different meteorological forcing on snowmelt that increases the rate of water percolation due to higher solar radiation loading (Fig. 2c). This increased radiation loading will also increase the rate of equilibrium snowpack metamorphism (to rounded grains) causing the snowpack to be less stratified relative to the flat or north aspect snowpacks. Results of this study suggest that meltwater is flowing at the SSI and

downslope through the snowpack on the north aspect hillslope but not on the south aspect hillslope.

The north aspect slope has preferential flowpaths during snowmelt that are similar to an alpine catchment with water flowing through the snowpack downslope (Liu et al., 2004, Williams et al., 2000), during rain on snow events at lower



elevation sites (Eiriksson et al., 2013), and observations in a coastal climate (Kattelmann and Dozier, 1999). The soil, snow, topographic characteristics, and rate of snowmelt create an environment on the north aspect slope that separates vertically percolating meltwater into flowpaths that travel across the SSI and downslope (Fig. 4) increasing SWE at the toe of the slope (Fig. 3) and allowing some water to infiltrate across the SSI similar to a permeability barrier (Fig.

5bii) (Webb et al., in review). The south aspect slope has coarser soil, less stratified snow, and higher rates of snowmelt due to increased solar radiation that produce less observed flow laterally at the SSI, and more infiltration into the soil. These processes can be combined into a conceptualization of the northern aspect slope having meltwater flowpaths near the SSI downslope and the southern aspect slope having more infiltration into the soil (Fig. 6). Coarse soil can also divert water in the form of a capillary barrier as observed by Eirikkson et al. (2015) at the rain-snow transition

zone, though this was not observed in our study. The slope of the hill will affect this phenomena (Webb et al., in review) and in this study the north aspect slope is steeper than the south aspect (Fig. 1) though further testing is necessary at multiple slope angles to investigate how slope controls this process, since both north and south slopes are steep enough to produce hydraulic barriers (Webb et al., in review).

As hydraulic barriers form and promote flowpaths to develop within the snowpack such as on the north aspect slope,

the timing of runoff at the hillslope scale can change dramatically. Snow has been shown to have a hydraulic conductivity orders of magnitude greater than common soils (Yamaguchi et al., 2010; Domine et al., 2013) and will thus be important for hydrologic modeling and flood prediction from snowmelt runoff. From a groundwater recharge perspective, much of the hydraulic gradients driving subsurface flow will be occurring at the base of the north aspect hillslope in this study area due to the lateral flow of water through the snowpack and soil moisture sensors on the slope

will only account for a fraction of the total meltwater as flowpaths bypass sensor profiles. Also, at the base of the hillslope (NT) the snowpack can increase in bulk SWE by up to 250 mm (from 146 mm to 396 mm, Fig. 3c) increasing the storage capacity of a location and resulting in areas of focused recharge and variable infiltration in the subsurface as observed in other subalpine regions (Webb et al., 2015). However, hillslopes can still display a more classical conceptualization of snowmelt infiltration uniformly and travelling across the soil-bedrock interface to recharge

groundwater resources and generate streamflow as on the south aspect slope (Fig. 6) and later in the summer on the north aspect slope (Fig. 5bii).

Preferential flowpaths and aspect controls during snowmelt has been observed at lower elevations. At a different site in Colorado, near the rain-snow transition zone, the intermittent snowpack on south aspects displayed matrix flow whereas north aspects displayed preferential flowpaths through the soil (Hinckley et al., 2014). These results are

similar to those observed in this study during the 2015 snowmelt while the 2013 and 2014 seasons displayed what can be interpreted as lesser melt rates on the north versus south slopes due to temperature and radiation differences (Fig 2c and d) that result in the preferential flowpaths at the SSI. In this study, the north aspect slope displays preferential flowpaths early in the snowmelt season similar to alpine regions (Liu et al., 2004; Williams et al., 2015) that can transport a large amount of water relative to the following melt periods (Fig. 3ai and bi). Flowpaths then transition to

more uniform melting and less preferential flow as the melt season progresses. The south aspect slope is similar to slopes at lower elevations near the rain-snow transition zone (Eiriksson et al., 2013; Hinckley et al., 2014) that display uniform melt and matrix type of flow with small amounts of water diversion at the SSI.





In 2013 the increase in SWE at the toe of the north facing slope was 30 mm greater than the amount of precipitation that was recorded (Fig. 3a). The increase in ΔSWE downslope is occurring from the accumulation of meltwater flowing across the SSI (Fig. 6). Wind is not likely causing increased deposition on any particular part of a hillslope since winds run perpendicular to slopes (Fig. 2). Though snow drifts may still occur, care was taken during

measurements to avoid areas with noticeable wind drifts, or where drifts would likely occur due to the predominant wind directions. In 2014, the large increase in SWE at the toe of the north aspect slope (NT) is from the lateral flow of water in snow and the rising of the water table above the soil surface (Fig. 3bi). This is a result of snowmelt primarily influencing the top 10 cm of soil on the slope (Blankinship et al., 2014) and water flowing downslope at the SSI decreasing the travel time of water on the hillslope and increasing connectivity at the toe of the hillslope and water

table similar to observations in the northern Rocky Mountains (Jencso et el., 2009). Some locations on the north aspect slope in 2014 remained consistent in the amount of bulk SWE as other locations on the hillslope decreased in SWE due to preferential flowpaths causing non-uniform flow across the hillslope (Fig. 4). The final 2015 survey shows similar increases in SWE between surveys that are the result of rain-on-snow events occurring that are known to produce lateral flow within snowpacks (Eiriksson et al., 2013). It is also difficult to determine if this would be isolated

to the north aspect hillslope in 2015 since there is a lack of snow on the south facing slope during the rain-on-snow events.

In flat terrain, snowmelt patterns are known to have correlation lengths of five to seven meters in alpine environments (Sommerfeld et al., 1994; Williams et al., 1999) and two to four meters in a subalpine environment (Webb, in review). These correlation lengths are less than the distances between measurement locations in this study. However, these

correlation lengths are explained by flow across snow layer interfaces and snow topography in flat terrain (Sommerfeld et al., 1994; Williams et al., 1999; Williams et al., 2010). Increasing the topographic slope will thus increase the correlation lengths as the snow layer interfaces tilt with the slope of the ground (Webb et al., in review). This study shows that the resulting correlation lengths in complex terrain with steep slopes can increase towards the scale of the terrain variability and result in increases in SWE at the toes of hillslopes. Further investigations are necessary to

determine the scale that water may flow through snow or at the SSI on steep slopes. Future studies will benefit from the use of numerous soil moisture sensors to obtain time-series data of soil VWC at multiple locations within a watershed to observe the variable infiltration characteristics during snowmelt that is difficult to detect from the near surface soil moisture.

When considering dynamic hydrologic processes that occur during spring snowmelt in subalpine headwater

catchments, it is important to consider the variable flowpaths that develop based on factors such as slope, aspect, soil parameters, and snowpack characteristics to move beyond single point measurements and one-dimensional assumptions. The toe of a hillslope is an important location to observe and estimate the amount of hillslope runoff occurring near or above the SSI relative to flow through the soil in future investigations. Future studies will benefit from considering the snowpack as an extension of the vadose zone during spring snowmelt due to the variable

saturated flow that occurs.





## 5 Conclusions

The observations of this study occurred during above normal, relatively normal, and below normal snow seasons capturing bulk SWE and soil VWC variability in space and time during spring snowmelt with varying meteorological forcing conditions, including rain-on-snow events in 2015. Evidence was presented of preferential meltwater flowpaths at the snow-soil interface on the north aspect hillslope during early snowmelt. The effect of these preferential flowpaths were observed in changes in SWE and infiltration in the shallow subsurface at 20 cm depth, but not observed in the near surface soil moisture. Near surface soil moisture is correlated the strongest to soil moisture measured during the first survey than to other topographic parameters or hydrologic variables. Infiltration beyond the near surface occurred more on flat terrain when compared to sloped conditions during the entire snowmelt season, resulting in greater relative saturation in the shallow subsurface in the flat area.

The snowpack is a porous medium that is an extension of the vadose zone and increases the water storage capacity of a region within a watershed. Water flowing downslope at the snow-soil-interface increased SWE at the toe of the north aspect hillslope by as much as 250 mm (170 %) that additionally effects the soil moisture at the toe of the slope. The south aspect hillslope did not display evidence of this phenomenon. The differences in flowpath development on the two opposite facing hillslopes is due to differences in soil, snowpack characteristics, slope and aspect, and snowmelt rates as a result of meteorological forcing variability. The formation of a hydraulic barriers at the snow-soil interface will be dependent upon both the snow and soil characteristics and conditions during melt. During 2015 when a relatively low peak SWE occurred early and rain-on-snow events were observed, the variability of snow and soil moisture increased displaying the connection and interactions between snow and soil moisture. Results from this study show that the snow acts as an extension of the vadose zone during spring snowmelt and future investigations will benefit from studying both the snow and soil together.

## 6 Acknowledgements

The authors would like to acknowledge multiple people that assisted with the work that is presented in this manuscript. The Colorado Ground Water Association provided financial assistance through the Harlan Erker Memorial Scholarship that was used to purchase the soil moisture and temperature sensors and datalogger. The Colorado State University snow hydrology field methods course (WR575) provided an abundance of field work assistance with additional volunteers from the Fassnacht snow laboratory and Sarah Schmeer was a helpful field assistant on a number of surveys. The Routt National Forest United States Forest Service was very helpful and provided valuable assistance for the research permitting process. Additionally, Dr. Jorge Ramirez and Dr. Jeffrey Niemann provided feedback on an earlier version of this manuscript that greatly improved the quality of this work. The authors express great appreciation for all those involved in the presented work.

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





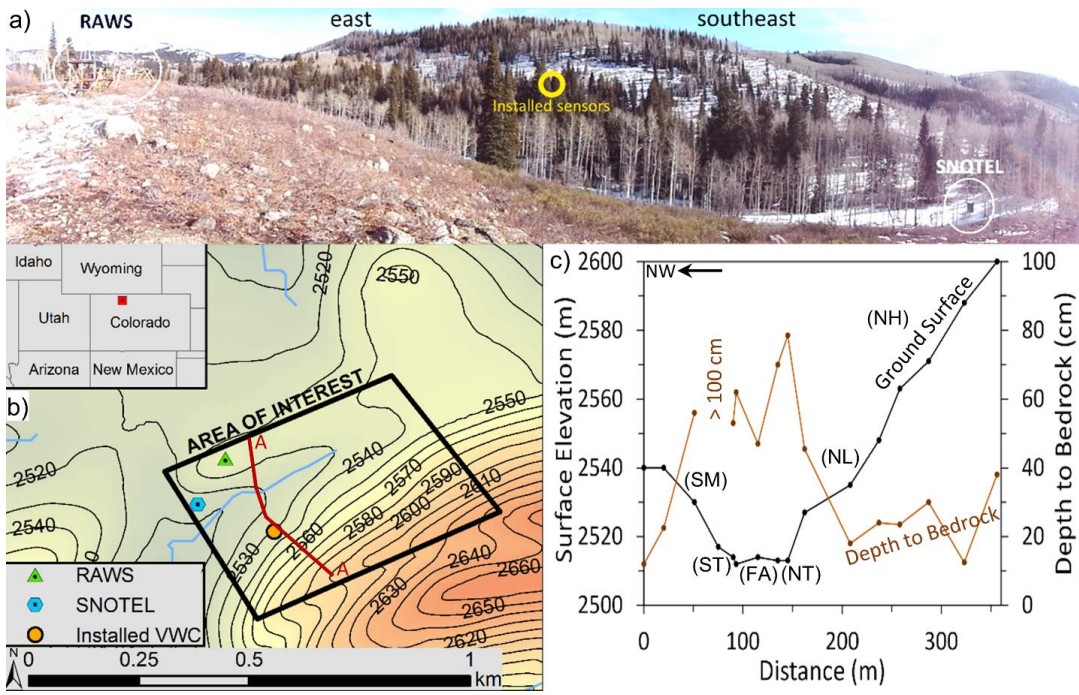

**Figure 1: a) Panoramic picture of the study area facing east to southeast. Location of photo taken to the west of RAWS location in map
(b). Locations of the Remote Automated Weather Station (RAWS), SNOTEL station, and installed soil moisture sensors are circled and
labeled. b) Map of the study site and area of interest in this investigation. 10 m contours are shown. c) Cross section A-A from panel (b)
showing the elevation of the ground surface and depth to bedrock using a 100 cm long hand auger. Regions of interest are identified as
middle of the south aspect hillslope (SM), toe of the south aspect slope (ST), flat aspect (FA), toe of the north aspect slope (NT), low on
the north aspect slope (NL), and high on the north aspect slope (NH). All ground surface data are 10 m resolution digital elevation model
(USGS, 2015).**

25





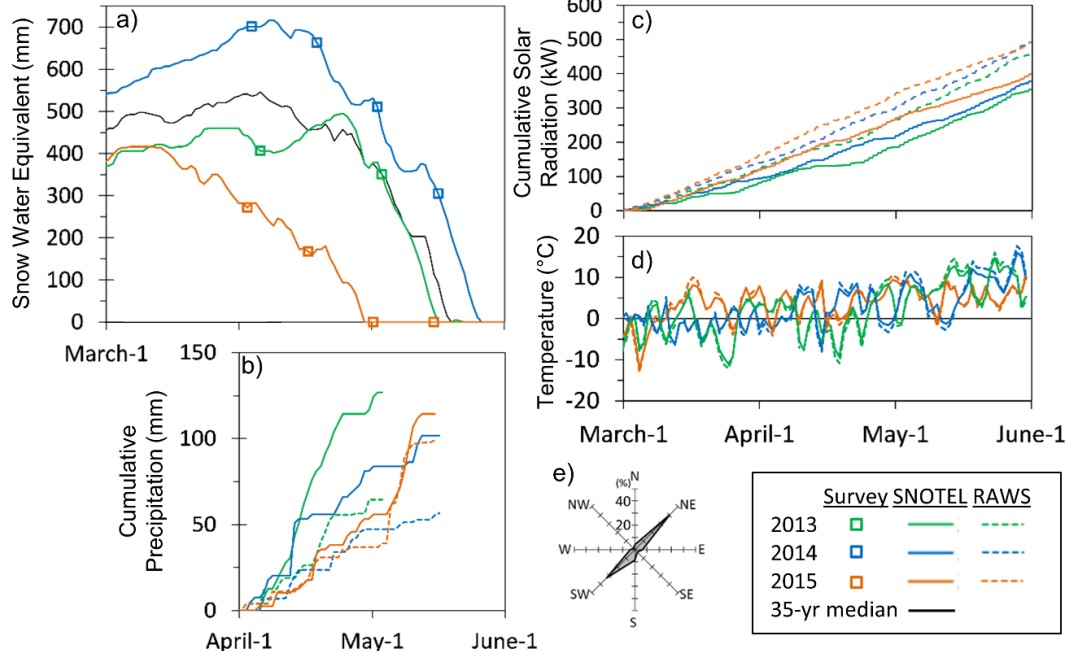

**Figure 2: a) Daily Snow Water Equivalent (SWE) measured at the SNOTEL station each spring of the study period and the 35 year median of the station measurements, b) cumulative precipitation occurring during the spring survey study periods of April and May as measured at the SNOTEL and RAWS site, c) cumulative solar radiation at the SNOTEL and RAWS sites during spring, d) mean daily temperature at the SNOTEL and RAWS sites during spring, and e) wind rose of spring data for the three years studied at the SNOTEL site.**





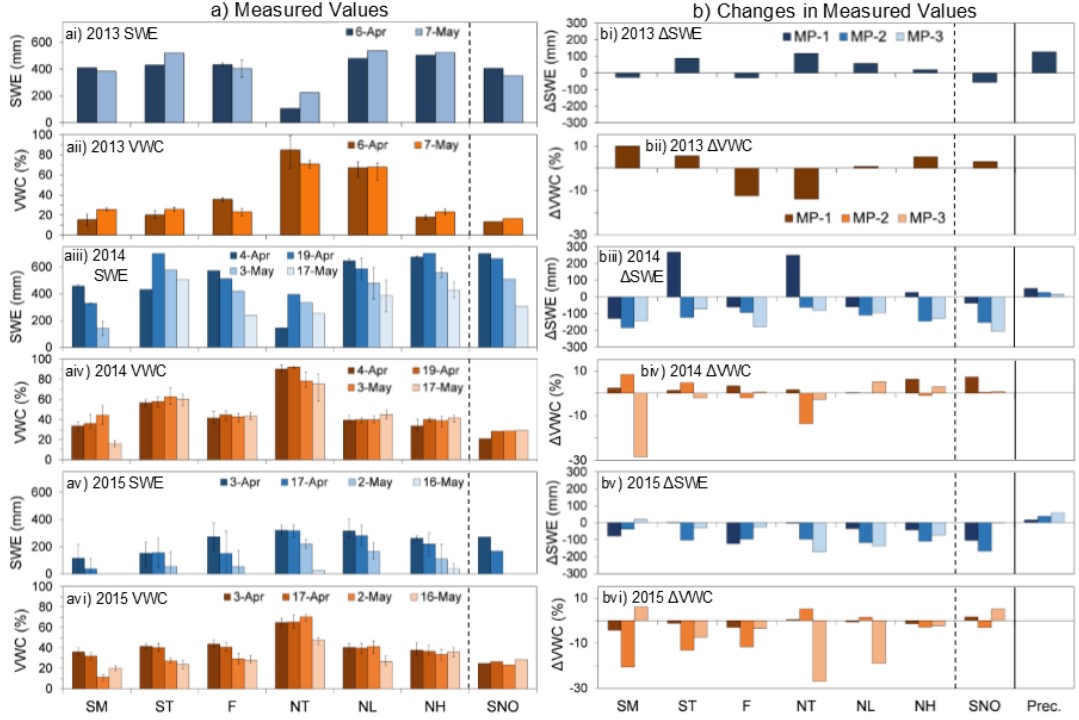

**Figure 3: Observations of a) measured values for snow water equivalent (SWE) and near surface soil volumetric water content (VWC) and b) changes in measured values between survey dates for regions of interest: middle of the south aspect hillslope (SM), toe of the south aspect slope (ST), flat aspect (FA), toe of the north aspect slope (NT), low on the north aspect slope (NL), and high on the north aspect slope (NH). Also included are observed values at the SNOTEL site (SNO) that include SWE and precipitation (Precip.). Figure panels display i) 2013 SWE, ii) 2013 VWC, iii) 2014 SWE, iv) 2014 VWC, v) 2015 SWE, and vi) 2015 VWC. Changes are shown for melt period 1 (MP-1) between the first two surveys, melt period 2 (MP-2) between the second and third surveys, and melt period 3 (MP-3) between the third and fourth surveys. Each melt period is 14-15 days with the exception of 2013 that was 28 days. Error bars indicate total range of measurements at locations.**





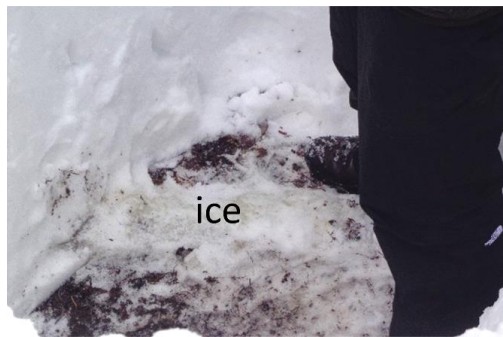

**Figure 4: Picture of frozen ice "vein" observed at the snow-soil-interface (SSI) with foot shown for scale.**





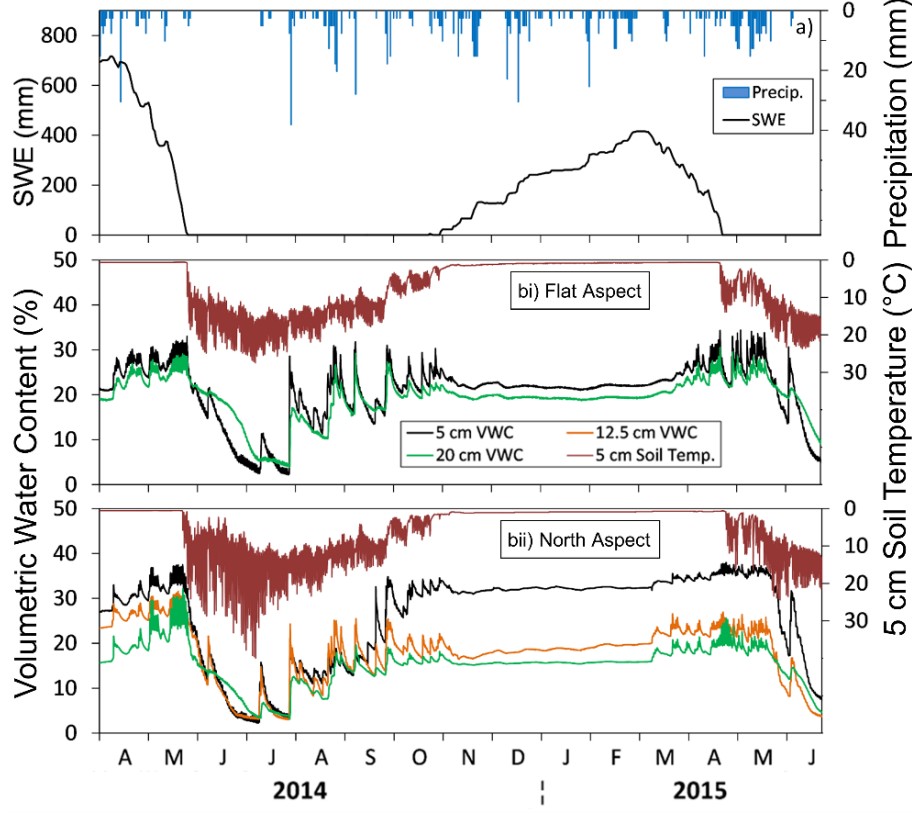

**Figure 5: a) Daily snow water equivalent (SWE) and precipitation recorded at the Dry Lake SNOTEL station and b) the hourly soil volumetric water content (VWC) and 5 cm deep temperature at bi) the flat SNOTEL site and bii) the installed sensors on the north aspect slope.**



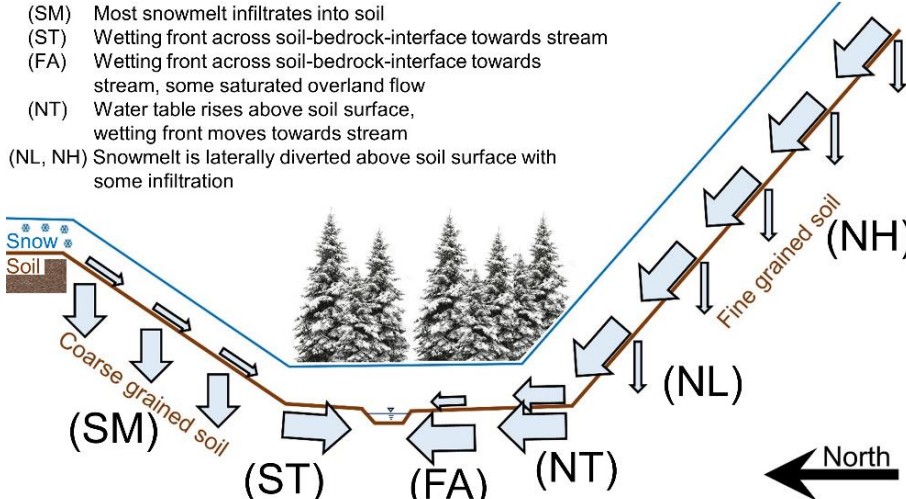

**Figure 6: Conceptual model of flowpaths that develop during early spring snowmelt at the south aspect hillslope (SM), toe of south aspect slope (ST), flat aspect (FA), toe of north aspect slope (NT), low on the north aspect hillslope (NL), and high on the north aspect hillslope (NH).**

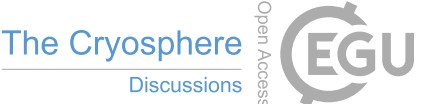



**Table 1: Percent of grain sizes by mass determined from sieve analysis of samples collected using a ~200 cm³ sample at locations in the middle of the south aspect slope (SM), the toe of the south aspect slope (ST), flat aspect (FA), and low on the north aspect slope (NL). Fines are considered less than 0.074 mm, sand is larger than fines and less than 4.75 mm.**

|                | SM | ST | FA | NL |
|----------------|----|----|----|----|
| **Percent Fines**  | 21 | 25 | 29 | 29 |
| **Percent Sand**   | 46 | 61 | 64 | 58 |
| **Percent Larger** | 33 | 14 | 7  | 13 |








**Table 2: Results of pattern analysis of near surface soil moisture measurements based on slope angle, northness (north.), elevation (elev.), Snow Water Equivalent (SWE), change in SWE (ΔSWE), SWE on first survey of the year, and near surface soil moisture (VWC) on first survey of the year. Significance is shown with table cell shading and bold text representing a p-value less than 0.05 and underlined text a p-value less than 0.01.**

| | | slope | north. | elev. | SWE | ΔSWE | 1st SWE | 1st VWC |
|---|---|---|---|---|---|---|---|---|
| **2013** | 6-Apr | 0.19 | 0.327 | 0.02 | **0.52** | ---- | ---- | ---- |
| | 4-May | 0.12 | 0.090 | 0.12 | -0.25 | -0.68 | -0.57 | **0.92** |
| **2014** | 4-Apr | -0.21 | 0.006 | -0.12 | -0.36 | ---- | ---- | ---- |
| | 19-Apr | -0.40 | 0.070 | 0.03 | -0.08 | **-0.82** | **-0.83** | **0.99** |
| | 3-May | -0.63 | 0.410 | -0.02 | 0.30 | -0.38 | **-0.89** | **0.95** |
| | 17-May | 0.57 | 0.494 | -0.40 | 0.62 | -0.48 | -0.39 | **0.82** |
| **2015** | 3-Apr | -0.18 | 0.181 | -0.22 | 0.12 | ---- | ---- | ---- |
| | 17-Apr | -0.08 | 0.262 | -0.01 | **0.56** | -0.09 | **0.47** | **0.95** |
| | 2-May | 0.11 | 0.015 | 0.26 | **0.87** | 0.36 | **0.56** | **0.80** |
| | 16-May | 0.06 | **0.417** | 0.26 | **0.46** | **0.50** | 0.15 | **0.48** |