# Peer review of "Hydrologic Flowpath Development Varies by Aspect during Spring Snowmelt in Complex Subalpine Terrain"

_The Cryosphere, 2017_

## Referee Comment (RC1) · Anonymous Referee #1 · 10 May 2017

In this paper, authors measured temporal changes of soil water content and snow water equivalent to estimate the influences of surrounding topography. This study is important to estimate lateral flow in the snowpack and at the snow soil interface. It is expected that this study estimate the amount of lateral flow under the provided topography, snow and meteorological conditions. In my opinion, analysis and discussion in this paper remains qualitative discussion without quantitative estimation. Perhaps, quantitative analysis is performed in another paper referred in the manuscript as Webb et al. (in review). If so, division of the paper reduces the impact of this paper. Even so, quantitative analyses from the observation results are necessary. For example, if the ratio of water for lateral flow and infiltration into the soil is added in Fig. 6 based on the result of

observation, this paper can provide scientific valuable information. In this paper, lateral flow is considered to be main cause of the increase of SWE at the toe. However, I think the authority of this consideration is not sufficient. If there are other evidence of large lateral flow (e.g. water saturated layer observed by snow pit observation), it should be shown.

minor comments

In introduction, one of the objectives is to investigate if preferential flowpath in snow can be observed in P3 L24. So observation of preferential flow should be described in detail in the body text.

P4 L25 According to this sentence, authors measured snow pit. Did authors measure not only SWE but also profiles of snow density and grain size? Depending on grain size and density profile, capillary barriers formed at the boundary of snow layers and provides lateral flow in the snowpack. Ice layers also provide lateral flow. If snow profiles are measured, observed result should be shown and used in the discussion of the lateral flow. Also, measuring method of SWE should be described. In this manuscript, lateral flow at snow-soil interface is considered main cause of the increase in SWE. So I guess the snowpack at the toe includes large water saturated layer. But water saturated layer make accurate SWE measurement more difficult. Therefore, more detailed description for observation is necessary.

P7L6 Ice veins were not observed in 2015. Is there a significant difference between 2014 and 2015? Comparison between 2014 and 2015 may suggest the influence of ice vein for lateral flow.

P8 L33-36 The last, "where observations of ice veins were made at the SSI.". Do authors think that the cause of lower water content at 20 cm depth in north facing slope is the existence of ice vein? But ice veins were not observed in 2015 according to P7 L6. I think that the slope affects water content. Lateral flow in soil prevents deep infiltration and leads to small water content at the depth area in north aspect slope.

P9 L14 Is Fig 4 a reference mistake of Fig 3? In the body text, Fig. 4 is sometimes referred in unnatural context (e.g. P10L3).

P9 L20-22 Is this discussion for only in 2014 or both in 2014 and 2015?

P9 L25 Fig. 4 is just a photo of ice vein. Do authors mean that this ice vein produces lateral flow? If so, it should be written in the caption of Fig. 4.

P9 L28-31 As this sentence show, increasing snow depth with SWE means less certainty of lateral flow. So figures of snow depth and discussion with snow depth are necessary. Also, if the snow profile was observed by pit observation, details of snow density profile should be shown.

P9 L32-34 Is this less stratified snowpack due to strong radiation in south facing slope confirmed by pit observation?

P9 L34-35 In author's opinion, which is the main reason of large lateral flow in north facing slope, ice vein, stratified snowpack or small hydraulic conductivity of soil?

P10 L20 Increase of 250 mm in SWE is too large to consider caused by lateral flow. Change in snow depth should be also shown to discuss.

P10 L32 Did the author observed preferential flow visually? In this paper, preferential flow formed in early snowmelt season on north facing slope and south facing slope displayed matrix flow in this time. However, preferential flow may form on south facing slope in earlier period before this observation. Snowmelt starts earlier on south facing slope than on north facing slope.

P11 L1-7 In this paragraph, cause of increase in SWE is considered lateral flow. But the authority of this seems insufficient. Especially, increase of 250 mm in 2014 is too large even if the effect of rising water table is considered. I wonder this discussion is based on the confirmation by snow pit observing large water ponding layer. Therefore, discussion about the reason why the cause of SWE increase is lateral flow is necessary. Description of "Wind is not likely causing increased deposition on any particular

part of a hillslope since winds run perpendicular to slopes." is not sufficient to determine lateral flow as cause. Explanation of confirmed fact by observation and authority of the cause are necessary.

P11 L12 Is Fig.4 reference mistake of Fig. 3?

―――――――――――――――――――

---

## Referee Comment (RC2) · Anonymous Referee #2 · 15 May 2017

In the present paper, the authors study the influence of preferential flowpaths in a deep snowpack on soil moisture at the hillslope scale and investigate notably its spatial variability. They perform a qualitative analysis of the SWE and soil moisture evolution between surveys and a statistical analysis (pearson's correlation coefficient) of near surface soil volumetric water content and topographical (aspect, slope), hydrological (peak SWE, date of peak SWE) variables. One of their main results is the increase in SWE observed at the beginning of the melt season at the toe of the north aspect hillslope. According to the authors, this increase is due to lateral flow in the snowpack. Even if this hypothesis is probable, the arguments provided in this study remains qualitative (observed ice veins at the snow-soil interface) and are not robust enough for

such a scientific publication. The introduction is well written and documented even if it is sometimes difficult to see where the authors want to go. In my opinion, some of the observations presented in the methods are the first weakness of this study: the authors present (and measured?) only the total SWE for each snow pit. A detailed profile with snow density and depth would be very useful in order to support the results (lateral flow as the main cause of SWE increase). Indeed, an in-depth comparison of snow layers could be very useful. The Results part is very difficult to read as it is essentially a description of figures: there is no underlying theme and the different paragraphs have few connections. The discussion is pretty good but does not bring any new argument. Finally, the authors are citing throughout the document their own paper currently under review (Webb et al., in review) but as a reviewer, the significance of this reference remains hard to assess.
* * *

---

## Author Comment (AC1) · 8 Jun 2017

The authors would like to first thank you for taking the time and effort to review our manuscript. We appreciate the constructive comments that have been made and our replies can be found below following a reiteration of the original comment.

In this paper, authors measured temporal changes of soil water content and snow water equivalent to estimate the influences of surrounding topography. This study is important to estimate lateral flow in the snowpack and at the snow soil interface. It is expected that this study estimate the amount of lateral flow under the provided topography, snow and meteorological conditions. In my opinion, analysis and discussion in this paper remains

qualitative discussion without quantitative estimation. Perhaps, quantitative analysis is performed in another paper referred in the manuscript as Webb et al. (in review). If so, division of the paper reduces the impact of this paper. Even so, quantitative analyses from the observation results are necessary. For example, if the ratio of water for lateral flow and infiltration into the soil is added in Fig. 6 based on the result of observation, this paper can provide scientific valuable information. In this paper, lateral flow is considered to be main cause of the increase of SWE at the toe. However, I think the authority of this consideration is not sufficient. If there are other evidence of large lateral flow (e.g. water saturated layer observed by snow pit observation), it should be shown.

Response: The other paper that is in review is a study at a different location and dataset with relevant results. However, we will certainly expand our citations to include other papers that we cited in the introduction. Given your comments, we agree that further quantification is justified and can certainly be addressed by adding an estimated ratio of lateral flow vs. infiltration to enhance this paper. We propose that an energy balance estimate of meltrates and accumulation along the hillslope required to produce the observed increases in SWE would provide an estimate of this ratio in revisions.

minor comments

In introduction, one of the objectives is to investigate if preferential flowpath in snow can be observed in P3 L24. So observation of preferential flow should be described in detail in the body text.

Response: Thank you for pointing this out. We should clarify this objective to say "evidence" of lateral flow can be observed at the hill slope scale since it is rarely investigated at this scale. We will directly address the re-phrasing of the objective and adding text to the main body of results to better fulfill this objective.

P4 L25 According to this sentence, authors measured snow pit. Did authors measure not only SWE but also profiles of snow density and grain size? Depending on grain

size and density profile, capillary barriers formed at the boundary of snow layers and provides lateral flow in the snowpack. Ice layers also provide lateral flow. If snow profiles are measured, observed result should be shown and used in the discussion of the lateral flow. Also, measuring method of SWE should be described. In this manuscript, lateral flow at snow-soil interface is considered main cause of the increase in SWE. So I guess the snowpack at the toe includes large water saturated layer. But water saturated layer make accurate SWE measurement more difficult. Therefore, more detailed description for observation is necessary.

Response: Snow profiles of density and grain size were not measured at all locations due to time constraints of an individual surveyor collecting data at all locations (often more than 10) within a single day. We did, however, collect these data at a single location on the first survey date of each year and are happy to include these data in revisions. We understand that capillary barriers as well as permeability barriers can cause lateral flow and have observations of ice lenses and saturated snow zones that can be further discussed for clarity, though the wet snow layer is mentioned on P7 L8. We will add further discussion to these points in revisions. The measuring method of SWE is described on P4 L31.

P7L6 Ice veins were not observed in 2015. Is there a significant difference between 2014 and 2015? Comparison between 2014 and 2015 may suggest the influence of ice vein for lateral flow.

Response: The main difference between 2014 and 2015 was the timing of peak SWE and higher temperature in 2015. In 2015 no increases in SWE downslope were observed. We will further discuss this point in revisions.

P8 L33-36 The last, "where observations of ice veins were made at the SSI.". Do authors think that the cause of lower water content at 20 cm depth in north facing slope is the existence of ice vein? But ice veins were not observed in 2015 according to P7 L6. I think that the slope affects water content. Lateral flow in soil prevents deep

infiltration and leads to small water content at the depth area in north aspect slope.

Response: We agree with this point. The presence of ice veins shows evidence that lateral flow above the SSI is occurring and that lateral flow in the top layer of soil is also contributing to the lateral flow downslope. We will revise the text to clarify this point.

P9 L14 Is Fig 4 a reference mistake of Fig 3? In the body text, Fig. 4 is sometimes referred in unnatural context (e.g. P10L3).

Response: You are correct. We apologize for this mistake and will correct it.

P9 L20-22 Is this discussion for only in 2014 or both in 2014 and 2015?

Response: This is discussion that combines observations from all years to form the conceptual model in Fig. 6. We will revise the text to clarify this.

P9 L25 Fig. 4 is just a photo of ice vein. Do authors mean that this ice vein produces lateral flow? If so, it should be written in the caption of Fig. 4.

Response: Language will be added to the caption. No, we do not believe that the ice vein produces lateral flow, but is rather evidence of lateral flow occurring.

P9 L28-31 As this sentence show, increasing snow depth with SWE means less certainty of lateral flow. So figures of snow depth and discussion with snow depth are necessary. Also, if the snow profile was observed by pit observation, details of snow density profile should be shown.

Response: This was the only location that depth also increased. We have the depth and bulk density information that will be added to help clarify. Though snow profiles at all pit locations were not collected on each survey date, detailed profiles were collected on the first survey dates each year and will be added.

P9 L32-34 Is this less stratified snowpack due to strong radiation in south facing slope confirmed by pit observation?

Response: It is confirmed through observations of ice lenses but not through detailed profile measurements. Additionally, data from other studies that show North aspect slopes to have more stratified snowpacks as a result of solar radiation loading in this region.

P9 L34-35 In author's opinion, which is the main reason of large lateral flow in north facing slope, ice vein, stratified snowpack or small hydraulic conductivity of soil?

Response: It is a combination, but the main reason would be the north facing slope causing lesser meltrates and low hydraulic conductivity of the soil. Just to be clear, the ice veins are a result of lateral flow, not the other way around. This will be clarified.

P10 L20 Increase of 250 mm in SWE is too large to consider caused by lateral flow. Change in snow depth should be also shown to discuss.

Response; Snow depth decreased at this location. It is also a result of the rising of the water table as discussed on P11 L7.

P10 L32 Did the author observed preferential flow visually? In this paper, preferential flow formed in early snowmelt season on north facing slope and south facing slope displayed matrix flow in this time. However, preferential flow may form on south facing slope in earlier period before this observation. Snowmelt starts earlier on south facing slope than on north facing slope.

Response: This is true. It is possible that preferential flow occurred during a time that observations were not being made. Here, we are comparing results to the Hinckley et al., 2014 study that is mentioned in the previous sentence. We are also making the argument based on the physics of flow through porous media. This will be further clarified and discussion of the possibility of pref. flow occurring on the S-Aspect earlier in the season added.

P11 L1-7 In this paragraph, cause of increase in SWE is considered lateral flow. But the authority of this seems insufficient. Especially, increase of 250 mm in 2014 is too
large even if the effect of rising water table is considered. I wonder this discussion is based on the confirmation by snow pit observing large water ponding layer. Therefore, discussion about the reason why the cause of SWE increase is lateral flow is necessary. Description of "Wind is not likely causing increased deposition on any particular part of a hillslope since winds run perpendicular to slopes." is not sufficient to determine lateral flow as cause. Explanation of confirmed fact by observation and authority of the cause are necessary.

Response: As mentioned in previous comments, we will provide more of the available data to further clarify these points. We also made a point to not measure locations that formed snow drifts behind trees, etc. and snow depth decreased as well as a deep saturated layers were observed.

P11 L12 Is Fig.4 reference mistake of Fig. 3?

Response: Yes, this is a mistake. Thank you for noticing this.

Again, I would like to reiterate our appreciation of the reviewer's constructive comments. Addressing these concerns will certainly improve the quality of the paper. Attached to this comment is a draft revision of the figure that shows the snow depth along with SWE.

[Figure]

**Fig. 1.**

---

## Author Comment (AC2) · 8 Jun 2017

The authors would like to first thank you for taking the time and effort to review our manuscript. We appreciate the comments that have been made and our responses can be found below immediately following a reiteration of the comment.

In the present paper, the authors study the influence of preferential flowpaths in a deep snowpack on soil moisture at the hillslope scale and investigate notably its spatial variability. They perform a qualitative analysis of the SWE and soil moisture evolution between surveys and a statistical analysis (pearson's correlation coefficient) of near surface soil volumetric water content and topographical (aspect, slope), hydrological

[Figure]

(peak SWE, date of peak SWE) variables. One of their main results is the increase in SWE observed at the beginning of the melt season at the toe of the north aspect hillslope. According to the authors, this increase is due to lateral flow in the snowpack. Even if this hypothesis is probable, the arguments provided in this study remains qualitative (observed ice veins at the snow-soil interface) and are not robust enough for such a scientific publication.

Response: We agree that only observations of ice veins would not be robust enough for publication in this journal. Points are made from multiple observations in the text and combined in the discussion based upon not only increases in SWE at both locations on the hillslope and at the toe of the hillslope (Fig. 3) in combination with the frozen "ice veins", but also less infiltration to 12.5 cm and 20 cm depths on the slope (Fig. 5), similarities in soil moisture between snowmelt and overland flow rain events, and the observations of snow density and saturated layers in the snow increasing downslope. We apologize for not making this clear in the writing and will revise to further highlight the multiple points of our argument. In revisions, we propose to further quantify the lateral flow by means of an energy balance model to estimate meltrates and calculate the required lateral flow accumulation along the hillslope necessary to produce the observed increases in SWE.

The introduction is well written and documented even if it is sometimes difficult to see where the authors want to go. In my opinion, some of the observations presented in the methods are the first weakness of this study: the authors present (and measured?) only the total SWE for each snow pit. A detailed profile with snow density and depth would be very useful in order to support the results (lateral flow as the main cause of SWE increase). Indeed, an in-depth comparison of snow layers could be very useful.

Response: We agree that a detailed snow density profile at all locations would be useful. However, due to constraints of an individual surveyor making observations at more than ten locations in a single day this was not feasible. The objective of this study was to gather more information spatially rather than at only a few points. However, we
are happy to add the depth observations to figures to help clarify that more data was collected than it seems the reviewer is aware of and we apologize for not making this more obvious in the writing. Furthermore, detailed profiles were collected on the first survey of each season and will be added.

The Results part is very difficult to read as it is essentially a description of figures: there is no underlying theme and the different paragraphs have few connections. Response: We apologize for this section being difficult to read. This is a writing style choice that the authors made where we are presenting only the results in the "results" section and produce the underlying theme and storytelling in the "discussion" section of the paper. We will revise the "results" section to better connect the paragraphs. The discussion is pretty good but does not bring any new argument.

Response: We believe that we do bring new arguments. No studies that we know of have combined snow observations with soil moisture observations at this scale and in this type of environment. Furthermore, our development of the conceptual model showing that water flow through snow is more important on the north aspect slope vs. south facing slopes as a result of snow and soil parameters has not been argued for in the literature to our knowledge.

Finally, the authors are citing throughout the document their own paper currently under review (Webb et al., in review) but as a reviewer, the significance of this reference remains hard to assess.

Response: We agree with this point entirely and will revise the manuscript by also citing fully published papers to support these points.

Again, thank you for your time in reviewing our manuscript.

---

## Author Comment (AC3) · 8 Jun 2017

I would like to first thank the editors and both of the reviewers for their time in working on our manuscript. The constructive comments provided by the reviewers will certainly improve the quality of the paper.

Both reviewers commented on the lack of detailed snow pit profiles at all of our observed locations. We agree that detailed snow density and grain size profiles at all locations would be useful. However, due to constraints of an individual surveyor making observations at more than ten locations, with some depths over two meters, in a single day this was not feasible. The objective of this study was to gather more in-

formation spatially rather than at only a few points. However, we will add the depth observations to figures to help clarify that more data was collected than it seems both reviewers are aware of and we apologize for not making this more clear in the writing. Detailed profiles that were collected on the first survey of each season and will be added and discussed along with observations of ice lenses and saturated snow layers during the following surveys.

Based on further reviewer comments we will additionally make the following revisions to the manuscript in addition to other minor comments made and responded to in the discussion board:

1. Quantify the ratio of lateral flow vs. infiltration by estimating meltrates through an energy balance model and calculating the required accumulation of lateral flow along the hillslope to produce the observed increases in SWE.

2. Add snow profile data collected on the first survey of each year and observations of ice lenses and saturated layers during other surveys along with further discussion of these observations.

3. Further clarify observations of changes in snow depth and add snow depth to figures.

4. Add further citations to support points made in the discussion as Webb et al., in review is still unpublished.

5. Modify objective to clarify what was being investigated for lateral flow and add wording in the main body to more directly address this objective.

6. Discuss potential for preferential flow to occur in 2015 prior to the observation period.

7. Further clarify the physical process of water flowing downslope in the top layer of soil in addition to within the snowpack.
* * *

---

## Author Response (AR1)

Dear Editors,

Thank you for your time in reviewing the manuscript entitled "Hydrologic Flowpath Development Varies by Aspect during Spring Snowmelt in Complex Subalpine Terrain" (MS No: tc-2017-12). We have strongly considered all points made by both reviewers and have edited the manuscript accordingly. Please find below our responses to each of the reviewers comments in *red italics*, including the changes made in the manuscript. I have also added as supplementary material a copy of the manuscript with all changes tracked to assist in the review of edits. Again, thank you.

Sincerely,

Ryan Webb

**Reviewer 1:**

In this paper, authors measured temporal changes of soil water content and snow water equivalent to estimate the influences of surrounding topography. This study is important to estimate lateral flow in the snowpack and at the snow soil interface. It is expected that this study estimate the amount of lateral flow under the provided topography, snow and meteorological conditions. In my opinion, analysis and discussion in this paper remains qualitative discussion without quantitative estimation. Perhaps, quantitative analysis is performed in another paper referred in the manuscript as Webb et al. (in review). If so, division of the paper reduces the impact of this paper. Even so, quantitative analyses from the observation results are necessary. For example, if the ratio of water for lateral flow and infiltration into the soil is added in Fig. 6 based on the result of observation, this paper can provide scientific valuable information. In this paper, lateral flow is considered to be main cause of the increase of SWE at the toe. However, I think the authority of this consideration is not sufficient. If there are other evidence of large lateral flow (e.g. water saturated layer observed by snow pit observation), it should be shown.
*Response:*

*The other paper that is in review is a study at a different location and dataset with relevant results. However, we will certainly expand our citations to include other papers. We agree that further quantification is justified and we addressed this by adding an estimated ratio of lateral flow vs. infiltration to enhance this paper. We conducted an energy balance estimate of meltrates and accumulation along the hillslope required to produce the observed increases in SWE.*

minor comments

In introduction, one of the objectives is to investigate if preferential flowpath in snow can be observed in P3 L24. So observation of preferential flow should be described in detail in the body text.
*Response:*

*Thank you for pointing this out. We have clarified this objective to say "evidence" of lateral flow can be observed at the hill slope scale since it is rarely investigated at this scale.*

P4 L25 According to this sentence, authors measured snow pit. Did authors measure not only SWE but also profiles of snow density and grain size? Depending on grain size and density

profile, capillary barriers formed at the boundary of snow layers and provides lateral flow in the snowpack. Ice layers also provide lateral flow. If snow profiles are measured, observed result should be shown and used in the discussion of the lateral flow. Also, measuring method of SWE should be described. In this manuscript, lateral flow at snow-soil interface is considered main cause of the increase in SWE. So I guess the snowpack at the toe includes large water saturated layer. But water saturated layer make accurate SWE measurement more difficult. Therefore, more detailed description for observation is necessary.

*Response:*

*Snow profiles of density and grain size were not measured at all locations due to time constraints of an individual surveyor collecting data at all locations (often more than 10) within a single day. We did, however, collect these data at a single locations during some of the survey dates and have now included these data. We understand that capillary barriers as well as permeability barriers can cause lateral flow and have observations of ice lenses and saturated snow zones that have been added and discussed further for clarity, though the wet snow layer is mentioned in the original manuscript on P7 L8. The measuring method of SWE is also described in the original manuscript on P4 L31.*

P7L6 Ice veins were not observed in 2015. Is there a significant difference between 2014 and 2015? Comparison between 2014 and 2015 may suggest the influence of ice vein for lateral flow.

*Response:*

*The main difference between 2014 and 2015 was the timing of peak SWE and higher temperature in 2015. In 2015 no increases in SWE downslope were observed. We have further discussed this point in the revisions.*

P8 L33-36 The last, "where observations of ice veins were made at the SSI.". Do authors think that the cause of lower water content at 20 cm depth in north facing slope is the existence of ice vein? But ice veins were not observed in 2015 according to P7 L6. I think that the slope affects water content. Lateral flow in soil prevents deep infiltration and leads to small water content at the depth area in north aspect slope.

*Response:*

*We agree with this point. The presence of ice veins shows evidence that lateral flow above the SSI is occurring. We agree that lateral flow in the top layer of soil is also contributing to the lateral flow downslope. We have revised the text throughout the document to say flow "near the SSI" to clarify this point.*

P9 L14 Is Fig 4 a reference mistake of Fig 3? In the body text, Fig. 4 is sometimes referred in unnatural context (e.g. P10L3).

*Response:*

*You are correct. We apologize for this mistake and have corrected it.*

P9 L20-22 Is this discussion for only in 2014 or both in 2014 and 2015?

*Response:*

*This is discussion that combines observations from all years to form the conceptual model in Fig. 7.*

P9 L25 Fig. 4 is just a photo of ice vein. Do authors mean that this ice vein produces lateral flow? If so, it should be written in the caption of Fig. 4.
*Response:*

*Language has been added to the caption. However, no we do not believe that the ice vein produces lateral flow, but is rather evidence of lateral flow occurring.*

P9 L28-31 As this sentence show, increasing snow depth with SWE means less certainty of lateral flow. So figures of snow depth and discussion with snow depth are necessary. Also, if the snow profile was observed by pit observation, details of snow density profile should be shown.
*Response:*

*This was the only location that depth also increased. We have now included the depth data to figure 4 to help clarify this. Though snow profiles at all pit locations were not collected on each survey date, observations of snow depth are discussed throughout.*

P9 L32-34 Is this less stratified snowpack due to strong radiation in south facing slope confirmed by pit observation?
*Response:*

*It is confirmed through observations of ice lenses but not through detailed profile measurements. Additionally, data from other studies show North aspect slopes to have more stratified snowpacks as a result of solar radiation loading in this region.*

P9 L34-35 In author's opinion, which is the main reason of large lateral flow in north facing slope, ice vein, stratified snowpack or small hydraulic conductivity of soil?
*Response:*

*It is a combination, but the main reason would be the north facing slope and low hydraulic conductivity of the soil. Just to be clear, the ice veins are a result of lateral flow, not the cause. This has been clarified in lines P10.*

P10 L20 Increase of 250 mm in SWE is too large to consider caused by lateral flow. Change in snow depth should be also shown to discuss.
*Response:*

*Snow depth decreased at this location as is now shown in figure 4. Furthermore our new energy balance analysis estimates this is the result of only 4% of meltwater being redirected laterally and remaining in the snowpack. It is also a result of the rising of the water table as discussed in the original manuscript on P11 L7.*

P10 L32 Did the author observed preferential flow visually? In this paper, preferential flow formed in early snowmelt season on north facing slope and south facing slope displayed matrix flow in this time. However, preferential flow may form on south facing slope in earlier period before this observation. Snowmelt starts earlier on south facing slope than on north facing slope.
*Response:*

*This is true. It is possible that preferential flow occurred during a time that observations were not being made. Here, we are comparing results to the Hinckley et al., 2014 study that is mentioned in the previous sentence. We are also making the argument based on the physics of flow*

*through porous media. This has been further clarified and discussion of the possibility of pref. flow occurring on the S-Aspect earlier in the season added.*

P11 L1-7 In this paragraph, cause of increase in SWE is considered lateral flow. But the authority of this seems insufficient. Especially, increase of 250 mm in 2014 is too large even if the effect of rising water table is considered. I wonder this discussion is based on the confirmation by snow pit observing large water ponding layer. Therefore, discussion about the reason why the cause of SWE increase is lateral flow is necessary. Description of "Wind is not likely causing increased deposition on any particular part of a hillslope since winds run perpendicular to slopes." is not sufficient to determine lateral flow as cause. Explanation of confirmed fact by observation and authority of the cause are necessary.
*Response:*

*As mentioned in previous comments, we have provided more of the available data to further clarify these points. We also made a point to not measure locations that formed snow drifts behind trees, etc. and snow depth decreased as well as a deep saturated layers were observed. Text further explaining these observations are now in the document.*

P11 L12 Is Fig.4 reference mistake of Fig. 3?

*Response:*

*Yes, this is a mistake. Thank you for noticing this.*

*Again, I would like to reiterate our appreciation of the reviewer's constructive comments. Addressing these concerns has certainly improved the quality of this manuscript.*

**Reviewer 2:**

In the present paper, the authors study the influence of preferential flowpaths in a deep snowpack on soil moisture at the hillslope scale and investigate notably its spatial variability. They perform a qualitative analysis of the SWE and soil moisture evolution between surveys and a statistical analysis (pearson's correlation coefficient) of near surface soil volumetric water content and topographical (aspect, slope), hydrological (peak SWE, date of peak SWE) variables. One of their main results is the increase in SWE observed at the beginning of the melt season at the toe of the north aspect hillslope. According to the authors, this increase is due to lateral flow in the snowpack. Even if this hypothesis is probable, the arguments provided in this study remains qualitative (observed ice veins at the snow-soil interface) and are not robust enough for such a scientific publication.

*Response:*

*We agree that only observations of ice veins would not be robust enough for publication in this journal. Points are made from multiple observations in the text and combined in the discussion based upon not only increases in SWE at both locations on the hillslope and at the toe of the hillslope (Fig. 3) in*

*combination with the frozen "ice veins", but also less infiltration to 12.5 cm and 20 cm depths on the slope (Fig. 5), similarities in soil moisture between snowmelt and overland flow rain events, and the observations of snow density and saturated layers in the snow increasing downslope. In our revisions we have additionally shown the lateral flow necessary to produce our observations is only 4% of melt.*

The introduction is well written and documented even if it is sometimes difficult to see where the authors want to go. In my opinion, some of the observations presented in the methods are the first weakness of this study: the authors present (and measured?) only the total SWE for each snow pit. A detailed profile with snow density and depth would be very useful in order to support the results (lateral flow as the main cause of SWE increase). Indeed, an in-depth comparison of snow layers could be very useful.

*Response:*

*We agree that a detailed snow density profile at all locations would be useful. However, due to constraints of an individual surveyor making observations at more than ten locations in a single day this was not feasible. The objective of this study was to gather more information spatially rather than at only a few points. However, we have added the depth observations to figures to help clarify. Furthermore, detailed profiles that were collected have been included and further discussion of specific observations at locations.*

The Results part is very difficult to read as it is essentially a description of figures: there is no underlying theme and the different paragraphs have few connections.

*Response:*

*We apologize for this section being difficult to read. This was a writing style choice that the authors made where we are presenting only the results in the "results" section and produce the underlying theme and storytelling in the "discussion" section of the paper. We have revised the "results" section and hope it reads better to you now.*

The discussion is pretty good but does not bring any new argument.

*Response:*

*We believe that we do bring new arguments. No studies that we know of have combined snow observations with soil moisture observations at this scale and in this type of environment. Furthermore, our development of the conceptual model showing that water flow through snow is more important on the north aspect slope vs. south facing slopes as a result of snow and soil parameters has not been argued for in the literature to our knowledge. In particular, at a site that does not have frozen soil.*

Finally, the authors are citing throughout the document their own paper currently under review (Webb et al., in review) but as a reviewer, the significance of this reference remains hard to assess.

*Response:*

*We agree with this point entirely and have added citations of fully published papers to support these points.*

*Again, thank you for your comments. We believe that your comments have helped us to improve upon the manuscript greatly.*

[revised manuscript text omitted]

---

## Author Response (AR2)

Dear Editor and Reviewers,

We would like to thank you for spending the time that you did in reviewing our revised manuscript. We appreciate the constructive comments that were given and believe that incorporation of these comments to our revisions has greatly improved the quality of the manuscript. We have addressed every comment. Please find below a restatement of each comment followed by our response in green text. We have also quoted the lines of the revised text to aid in reviewing our response. Following these comments is a copy of the revised manuscript with tracked changes. Again, thank you.

10 Sincerely,

5

Ryan Webb

- 15 Editor Decision: Reconsider after major revisions (10 Oct 2017) by Valentina Radic Comments to the Author: Dear authors,
- Your revised manuscript has by now been reviewed by both original reviewers. They both acknowledge that the quality of the manuscript has improved, however, they provided more comments that need to be addressed. Please find attached their reviews below.

Please address these comments point by point in your response letter and submit the responses together with your re-revised manuscript.

25

30

Thank you, Valentina

Review#1

In revised manuscript, it was recognized that most of responses and revisions for each comments were conducted appropriately.

Addition of snow depth in Figure 4 provided clear information that SWE increased despite snow depth decreased. I understand bulk snow density increased considerably at NT. As far as my interpretation

35 of Fig 4, bulk density at NT increased from 200 to 400 kg/m3 or so from first measurement to second one. Now it is understandable that lateral flow in the snowpack seems significant effect.

I felt that additional description is necessary about the estimation of ratio of lateral flow. As a response of quantitative analysis, ratio of lateral flow was estimated using calculated snowmelt

40 amount. I think this method is appropriate. On the other hand, I felt many points are unclear about this estimation. For example, how was the ratio calculated to 4%? How long period was used to estimate the value 4% (for whole winter or specific period)? Was there a difference of ratio of lateral flow between north and south facing slope? Adding these descriptions provide more available information.

45

Thank you for taking the time to review our manuscript a second time.

The ratio of 4% was calculated through contributing areas to the locations of the snow pits. We used the elevation DEM to make a approximation of this contributing area for the observation locations. As for the difference with the south facing slope, this calculation was not conducted. The main reason that this was not done is that little evidence exists that lateral flow occurred on the south facing slope and any lateral flow that did occur would drain from the snowpack rather quickly due to the soil properties of that hill slope inhibiting our ability to observe these changes in SWE. Below are the text quotes of the new manuscript.

**55**

**P11 L16-18**

"Using the 10 m DEM we estimated average contributing areas for the snow pit locations on the north facing slope. We then used the changes in SWE and observed precipitation to estimate the contribution of lateral flow for the two week periods between observations.

**Review#2**

General comment:

- 5 The Introduction and Methods sections were mainly kept unchanged. The Results and Discussion parts were partially modified. The additional data (density profile) presented in the Results part don't really bring any added value. The discussion is still redundant and several inconsistencies are still present. I would suggest a much more in-depth modification of the paper.
- 10 Thank you for taking the time to review our manuscript again. We have revised the text taking careful consideration of your comments and believe it has improved the quality of the paper. The reason that the density profiles were added is because it was a point made from both reviewers of the original manuscript; though we agree that little is added from the profile other than the context of snow layering to compare the years. It is pretty standard for snow studies and readers may be expecting to
- 15 see it, so we chose to leave it in.

We have also revised the text to reduce redundancy beyond only your specific comments, please see our responses below.

20 Detailed comments:

P3 L10 Not sure Webb et al. is appropriate at that place.

More generally, this reference hasn't been changed throughout the text (mix between Webb et al. in review and Webb et al., 2017).

- We believe that it is appropriate and the title of the paper in the references ("The The Presence of Hydraulic Barriers in Layered Snowpacks: TOUGH2 Simulations and Diversion Length Estimates") should help show how it is applicable here. We added the Avanzi reference to ensure that it was not only an "in review" paper that supported the claim. Unfortunately that journal took 1 year for the first round of reviews so it is still in review. We believe it will be published in the near future. Furthermore, the Webb et al., in review and Webb, 2017 are two separate references.
- 30
- Webb, R. W.: Using Ground Penetrating Radar to Assess the Variability of Snow Water Equivalent and Melt in a Mixed Canopy Forest, Northern Colorado, Frontiers of Earth Science, doi: 10.1007/s11707-017-0645-0, 2017.
- 35 Webb, R. W., Fassnacht, S. R., Gooseff, M. N., and Webb, S. W.: The Presence of Hydraulic Barriers in Layered Snowpacks: TOUGH2 Simulations and Diversion Length Estimates, Transport in Porous Media, in review.

P6 L8-11 Why are you not mentioning 2014 in your analysis?

40 The reason is that these statements are only discussing the range of earliest to latest peak dates. We have added text to clarify this point so readers are not confused.

P6 L8-11

"Peak SWE timing ranged from the earliest on March 9 in 2015, preceding first survey by nearly one
 month, to the latest on April 25 in 2013, 19 days after the first survey (Fig. 2a). The number of days
 from peak SWE to no snow recorded at the SNOTEL station ranged from the fewest in 2013 of 22 days
 to the most in 2015 of 52 days..."

P6 L11-13 What fraction of precipitation was recorded as snow/rain (based on temperature measurements)? This would be interesting to have an estimate and put it in relation with the SWE increase observed in Fig 4.
 Agreed. Please see revised fig. 4. below that now includes a red component of the precipitation bar that represents precipitation that fell when air temperature (Tair) was greater than 1°C.

- P6 L25-33 First, I would suggest a step chart instead of a line for the density graph. This would help
  the reading of the graph (and maybe highlighting relevant features). This paragraph sounds highly
  hypothetical ("that display indicators of melt-freeze crusts...", "likely as the result of melt-freeze
  cycles") given that the measured densities (Fig. 3) are not that high. In addition, I don't really
  understand the added value of those observations with regard to the main message. The profiles were
  done near the SNOTEL site, closer to the south aspect slope and should be homogenous (following
- 10 your argumentation). Could you please clarify this point? We agree with your first point. The figure is now a step graph. I do not follow why the profiles would be homogenous. Regardless of if lateral flow is occurring in snow there will still be metamorphism and compaction processes that create a stratified snowpack. The new figure is shown here.

25

P7 L5-8 As mentioned before, you could compare the change in SWE with the new precipitation and if possible, estimate which part of it due new snow.

20 This comparison is conducted in the energy budget analysis in the discussion and have been revised to clarify that this was being done.

**P11 L17-18**

"We then used the changes in SWE and observed precipitation to estimate the contribution of lateral flow for the two week periods between observations.

P7 L20-21 How can you affirm that ice veins are continuous? Please clarify.

We used the term "appear" because we can not affirm that they are continuous for the entire slope, we have added language to clarify.

The text now reads: P7 L21-22 "...appeared to be continuous, though continuity was only confirmed 5 for three to four meters based on excavation."

P7 L32 How do you define that the soil is super-saturated? We define this by observed ponding on the surface, we have now added this wording to be sure it is understood clarify.

P7 L 32-33"...super-saturated during a single survey on April 19, 2014 (resulting in the 85% VWC and surface ponding, Fig. 4bii)"

- P8 L31-33 The difference seems much smaller in 2014 (figure 6) between VWC at 5cm an 12cm. How 15 do you explain this? Wouldn't be relevant to show the entire year 2014? It may be a result of the mid-winter installation. We have also added wording to further explain this, though this was mentioned in the methods already to explain why the entire 2014 year was not shown. We have also edited the text throughout to highlight the difference at 20 cm depth with less 20
- emphasis on the 12.5 cm VWC.

P5 L29-31: "Installation in December 2013 required disturbing the snowpack and soil, thus the snowpack and soil moisture were allowed to return to near undisturbed conditions after installation and data prior to March 15, 2014 was not included in analysis."

25

40

10

P8 L 32-33: "This is more pronounced in 2015 where the 2014 season may have been impacted by the mid-winter installation of the sensors"

P9 L12-13 Can you really compare VWC values from a flat and sloped terrain like this? There are of 30 course differences in soil types and drainage properties (induced by the slope)! We agree. However, the point of this paper is that drainage is different on the slopes and the snowpack is part of the sloped drainage dynamics. The combination of all components including the

soil, slope, and snowpack are what create the observations and is the main point of our manuscript. We discuss all of these factors in great detail in the discussion, though the combined effects are 35 summarized in the lines.

~P12 | 30-32

"...it is important to consider the variable flowpaths that develop based on factors such as slope, aspect, soil parameters, and snowpack characteristics to move beyond single point measurements and one-dimensional assumptions."

P9 L20-23 From what you say (P7 L23: These ice "veins" were not observed in 2015) and show in Figure 6, these two observations (ice veins and VWC values) were not done the same year. Your sentence suppose the opposite and is misleading.

We agree and see your point. We were not intending to be misleading, the text has been revised in to 45 be more clear that there was less infiltration to 20 cm depth in 2014 during the early meltseason when the ice veins were observed.

P9 L21-23

50 ....with more infiltration wetting the soils at 20 cm depth on the flat aspect and lesser wetting at this depth on the north aspect in the early meltseason (Fig. 6b) where observations of ice "veins" and saturated layers of snow were made at the SSI (Fig. 5)."

P9 L25-26 Your statement is clearly not always correct (see in figure 4 for example aii) ST vs NH, aiv) NL or NH vs ST avi) NH vs ST). Please be more cautious. 55

Thank you for pointing this out, it was intended to pertain to locations on the slopes and not at the toes (SM, NL, and NH). You were comparing the ST location that was not intended as part of this statement. We agree that it was previously worded in a way that this was not clear and have revised it.

P9 L25-26

"...north aspect soils often having similar and/or higher water contents than south aspect soils, both with and without the presence of a snowpack (Fig. 4, SM vs. NL and NH)."

5

P9 L26-27 This sentence doesn't bring any interesting information. Consider removing it! We agree. The sentence has been removed.

P9 L31-36 How do you explain the negative correlation? Try to clarify that part of the discussion (P9 L31 to P10 L6).

This negative correlation means that lower peak SWE (during first survey) correlates to higher VWC later in the season. We have reworded the sentences to be more clear in our explanation.

P9 L32-36

15 "This negative correlation indicates that locations with lower peak SWE (During the first survey of the season) tend to have greater VWC at the later surveys. This is the result of the shallower snowpacks during the first survey being near the bottom of the slopes and in the flat terrain influenced by canopy interception (Fig. 1a) and the following melt flows downslope at the SSI towards these locations increasing both SWE and near surface VWC during the following surveys"

20

P10 L7-11 As mentioned above, you are misleading the reader by putting together observations from different years. Increase in SWE and ice veins was observed in 2013 and 2014, reduced infiltration in 2015 mostly.

We were not intending to be misleading and have added wording to be more clear. The reduced

25 infiltration to 20 cm depth does also occur in 2014. Particularly during the early meltseason when the ice veins were observed.

**P10 L7-8**

"...the frozen "ice veins" observed in 2013 and 2014 early melt seasons (Fig. 5), less infiltration to 20 cm depth on the slope (Fig. 6),..."

P10 L16 Not sure Webb et al. is appropriate here.

We believe that the title should speak to how it relates, though it will be more clear in its appropriateness when it is published. This is

Webb, R. W., Fassnacht, S. R., Gooseff, M. N., and Webb, S. W.: The Presence of Hydraulic Barriers in Layered Snowpacks: TOUGH2 Simulations and Diversion Length Estimates, Transport in Porous Media, in review.

40

30

P10 L16-17 How can you justify from figure 4 that not ice veins were observed on south facing slope? We see your point and have removed the reference to that figure. However, the evidence for lateral flow extends beyond ice veins to the increases in SWE (Fig. 4) and saturated layers within the snowpack that were not observed on south facing slope.

45

P10 L24-25 Why slower melt rates would induce lateral flow in the snowpack? Please clarify! This is a result of the physics of unsaturated flow. We have added a sentence to clarify the effects of slope and infiltration rate, and offer the appropriate references to refer the readers to the appropriate literature to learn more details about these physics. We feel that going into too much detail concerning

50 the physics of unsaturated flow would make the article longer than necessary. These processes are the primary focus of the entire paper Webb et al., in review.

~P10 L19-21

"When capillary barriers occur, the diversion length will be controlled by the hydraulic properties of the media, slope of the interface (steeper slope increases diversion length), and infiltration rate (slower infiltration rate increases diversion length) (Webb, 1997; Webb et al., in review)."

The paragraph P10 L29 to P11 L7 is mainly a repetition of elements already mentioned in paragraph (P10 L7 to L28). Revise this part of the discussion and remove redundancy. We agree and have compiled these two paragraphs into a single paragraph.

5

P10 L6-30:

"Meltwater flowing downslope near the SSI on the north aspect hillslope is shown by the increases in SWE at locations on and at the toe of the hillslope (Fig. 4), the frozen "ice veins" observed in 2013 and 2014 early meltseasons (Fig. 5), less infiltration to 20 cm depth on the slope (Fig. 6), similarities

- 10 in soil moisture between snowmelt and overland flow rain events, and the observations of snow density and wetness increasing with depth downslope in each north aspect snow pit. For the south aspect slope, the increases in SWE at the ST locations were similar to observed precipitation in 2013 and an increase in snow depth for 2014. The south aspect slope may have meltwater flowing downslope near the SSI, though to a lesser extent than the north facing slope and less apparent. The
- 15 movement of water across layer interfaces has been shown within a snowpack (Williams et al., 2000; Liu et al., 2004; Williams et al., 2010) and at the SSI (Eiriksson et al., 2013), with evidence of the latter being observed in this study (Fig. 5). This phenomenon will depend on soil parameters, snowpack layer characteristics, slope angle, and the rate that meltwater is percolating through the snowpack. These factors will determine if an interface acts as a permeability barrier, similar to a soil
- 20 drain, or a capillary barrier (Avanzi et al., 2016; Webb, 1997; Webb et al., in review). The primary reasons for lateral flow through the snowpack on the north facing slope is a result of the slower melt rates and hydraulic conductivity of the soil. When capillary barriers occur, the diversion length will be controlled by the hydraulic properties of the media, slope of the interface (steeper slope increases diversion length), and infiltration rate (slower infiltration rate increases diversion length) (Webb,
- 25 1997; Webb et al., in review). It is also possible for lateral flow to be caused by barriers within a layered snowpack well above the SSI, though the large saturated layer of snow was observed only at the SSI in all north aspect snow pits showing this is where the bulk of the lateral flow occurs. Further testing and field experiments are necessary to quantify the influence of varying slope and soil parameters on these processes in and below a snowpack. Our study shows preferential flowpaths
- 30 during snowmelt on the north facing slope that are similar to an alpine catchment with water flowing through the snowpack downslope (Liu et al., 2004, Williams et al., 2000), during rain on snow events at lower elevation sites (Eiriksson et al., 2013), and observations in a coastal climate (Kattelmann and Dozier, 1999).. These processes can be combined into a conceptualization of the northern aspect slope having meltwater flowpaths near the SSI downslope and the southern aspect slope having more infiltration into the soil (Fig. 7)."

P11 L15-16 You are referring to the wrong figure (Figure 4c) and consider revising the sentence "displaying the increased the storage capacity".

40 Revised.

**P11 L1-2**

"...increase in bulk SWE by up to 250 mm (from 146 mm to 396 mm, Fig. 4aiii) displaying the increased storage capacity of a location by the porosity of the snow."

45

50

P11 L20 Why are you saying that north slope can generate streamflow later in the summer (and referring to Figure 6bii)? Please clarify!

This figure shows the longer persistence of the snowpack later in the summer when there is no longer snow on the south facing slope or flat (SNOTEL). However, we agree that this may have been a confusing point and have thus removed this part of the sentence.

P11 L21-37 I am a bit skeptical that an increase of 170% in SWE can only explained by lateral flow in the snowpack. And how do you explain the increase at the toe of the south aspect slope? I would suggest that the rise of the water table is a major contributor to this change in SWE (as you mentioned of the discussion).

55 mentioned at the end of the discussion). Yes, we agree and explain that it is a contributor but increases in SWE upslope at NL where water table is not rising also confirm the lateral flow process and were used to determine the 4% calculations. South facing increases in SWE are at a location that is also increasing in depth from topographic variability so not necessarily increasing in density from lateral flow like the NT. When the south facing location does increase without a similar increase in depth it is close to the observed precipitation so difficult to separate. We have added text to clarify this difficulty at the south facing slope.

**5 P11 L16-25**

"Using the 10 m DEM we estimated average contributing areas for the snow pit locations on the north facing slope. We then used the changes in SWE and observed precipitation to estimate the contribution of lateral flow for the two week periods between observations. Given the observed increases in SWE on the north facing hillslope this results in a minimum of 4% of melt traveling

- 10 laterally above the SSI to produce the observed increases in SWE. The 4% is water that flows downslope above the SSI and remains in the snowpack. Therefore, the percentage may be larger when considering drainage from the snowpack after flowing laterally. Though 4% of melt flowing downslope within a snowpack is a low number it accumulates along the 250 m hillslope to increase the SWE at the toe of the slope the most. The 170% increase in SWE at NT observed in 2014 (Fig. 4) is
- 15 likely a result of both water flowing above the SSI and below it causing the water table to rise, though the increases in SWE at NL can be attributed to flow above the SSI."

**P10 L10-12**

"For the south aspect slope, the increases in SWE at the ST locations were similar to observed

20 precipitation in 2013 and an increase in snow depth for 2014. The south aspect slope may have meltwater flowing downslope near the SSI, though to a lesser extent than the north facing slope and less apparent."

P12 L4-7 I don't understand how you justify the different behaviors between 2015 and 2013-2014

25 (notably the link between melt and soil behavior). Just because of the shallower snowpack? Could you clarify this point please?

These lines do not relate to this, though we believe that the revised version explains our justifications. In 2015 our observation period was much after peak SWE and there was a number of rain events that created a different environment. In responding to your comments above we have clarified the reduced infiltration to 20 or double backward 2015. We have added the entropy below to

30 infiltration to 20 cm depth in both 2014 and 2015. We have also added the sentence below to recognize that we were unable to observe some of the processes in 2015 due to the early peak SWE.

**P12 L25-28:**

"Future hillslope scale investigations of these phenomena may benefit from larger scale runoff
 lysimeter studies similar to Eiriksson et al. (2013) and observing the entire melt season to capture peak SWE processes in low years."

**P12 L26 figure 53....**

Thank you for pointing this out. We have corrected it and given the revised document a solid proofread to catch any other mistakes similar to this.

P12 L28 You state: "The south aspect hillslope did not display evidence of this phenomenon". In 2013 and 2014, you observed an increase in SWE at the toe of the south facing slope (Figure 4 bi and biii). How do you explain it? I would lean towards a rise of the water table in the valley (encompassing ST F and NT sites) that would partly bias all measurements.

**45 and NT sites) that would partly bias all measurements. The water table was not observed to rise up into the snowpack at the ST location. Please also see above response concerning the explanation for the rise in SWE at ST.**

**P10 L10-12**

- 50 "For the south aspect slope, the increases in SWE at the ST locations were similar to observed precipitation in 2013 and an increase in snow depth for 2014. The south aspect slope may have meltwater flowing downslope near the SSI, though to a lesser extent than the north facing slope and less apparent."
- 55 In Figure 2a, modify the x-axis to have a tick every month. It does have a tick every month, though the box marker covers it up for May.

[revised manuscript text omitted]